# Self-Supervised Detection of Perfect and Partial Input-Dependent Symmetries

## Abstract

Group equivariance ensures consistent responses to group transformations of the input, leading to more robust models and enhanced generalization capabilities. However, this property can lead to overly constrained models if the symmetries considered in the group differ from those observed in data. While common methods address this by determining the appropriate level of symmetry at the dataset level, they are limited to supervised settings and ignore scenarios in which multiple levels of symmetry co-exist in the same dataset. For instance, pictures of cars and planes exhibit different levels of rotation, yet both are included in the CIFAR-10 dataset. In this paper, we propose a method able to detect the level of symmetry of each input without the need for labels. To this end, we derive a sufficient and necessary condition to learn the distribution of symmetries in the data. Using the learned distribution, we generate pseudo-labels that allow us to learn the levels of symmetry of each input in a self-supervised manner. We validate the effectiveness of our approach on synthetic datasets with different per-class levels of symmetries e.g. MNISTMultiple, in which digits are uniformly rotated within a class-dependent interval. We demonstrate that our method can be used for practical applications such as the generation of standardized datasets in which the symmetries are not present, as well as the detection of out-of-distribution symmetries during inference. By doing so, both the generalization and robustness of non-equivariant models can be improved. Our code is publicly available at url-removed-for-double-blind-review.

## 1 Introduction

Symmetry transformations change certain aspects of the world state (e.g. shape) while maintaining others unaffected or invariant (e.g. class). Introducing inductive biases into the model architecture that reflect the underlying symmetries of the data has progressively become a key principle in the design of more efficient neural networks (Higgins et al., 2018). This is often achieved through the use of equivariance, a property that guarantees that a certain transformation made to the input of a neural network will result in a equivalent transformation in the corresponding output.

Group equivariance leads to better generalization when the symmetries present in the data correspond to those in the group. However, if this is not the case, equivariance leads to overly constrained models and worse performance. To address this, common approaches involve manually adjusting the choice of the group to better reflect the symmetries in the data (Weiler & Cesa, 2019), or restricting the equivariance to subsets of the group. The latter is the case of Partial G-CNNs (Romero & Lohit, 2022), which implement partial equivariance layers and learn the subset $\mathcal{S} \subseteq \mathcal{G}$ that best represents the symmetries in the data. This avoids overly constraining the model, as the equivariance is respected only for the learned level of symmetry in the data. Importantly, Partial G-CNNs learn this level of symmetry in a supervised manner and at a the dataset level, which means that they are unable to recognize unique, input-specific levels of symmetry. This could pose a problem when different classes in the dataset exhibit varying symmetry levels.

In this paper, we introduce a technique for learning input-dependent levels of symmetry at a sample-level, without the need for labels. To achieve this, we build upon the Invariant-Equivariant Autoencoder (Winter et al., 2022), and infuse it with the ability to learn partial symmetries. We focus on the continuous group of rotations SO(2). Drawing parallels with Partial G-CNNs, we first explore uniformly distributed rotational symmetries $\mathcal{U}[-\theta, \theta]$ with a symmetry boundary $\theta$, and derive a

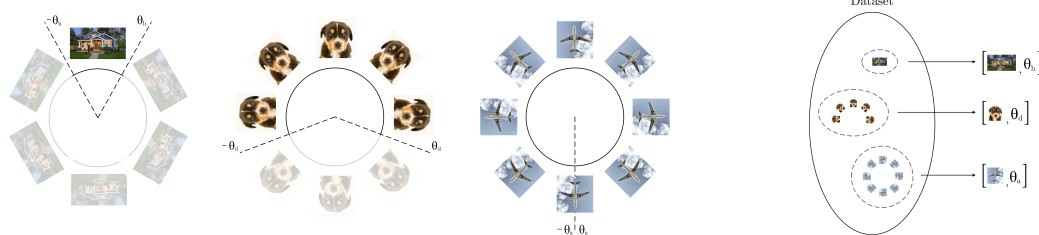

Figure 1: Self-supervised detection of input-dependent symmetries. In real world scenarios, different classes of objects present different levels of symmetries (*left*). Nevertheless, existing methods assume the same distribution of symmetries for all elements of the dataset. Our method can identify and determine the distribution of symmetries inherent to each input (*right*).

sufficient and necessary condition to learn the distribution of symmetries in the data. Next, we use the learned distribution to generate pseudo-labels, facilitating a self-supervised learning approach that allows a network to associate inputs with their specific symmetry levels. As a result, we are able to predict the levels of symmetry of unseen objects as well as detect out-of-distribution symmetries during inference. In addition, the proposed constraint on the group function can be used to reorient the inputs towards their centers of symmetry, allowing for generation of standardized datasets in which the symmetries are not present. In summary, our contributions are:

- We introduce a novel method to learn input-dependent levels of symmetries from data without the need for labels. Our method is able to determine both partial symmetries (subsets of the group) and perfect symmetries (spanning the entire group).
- We establish a sufficient and necessary condition to learn the subsets of symmetries in the data via group action estimators.
- We validate on both real and synthetic datasets the efficiency of our method in predicting input-dependent symmetry levels. Additionally, we present novel practical applications, including the detection of out-of-distribution symmetries and the generation of standardized datasets, which can be leveraged to improve the performance of non-equivariant models.
- We demonstrate that our framework is general enough to support a variety of symmetry distributions, such as arbitrary unimodal, symmetric distributions and discrete groups.

## 2 PRELIMINARIES

Our method builds upon the work of the Invariant-Equivariant Autoencoder (Winter et al., 2022) and æis motivated by Partial Group Equivariant CNNs (Romero & Lohit, 2022) ideas. In this section, we introduce these methods as well as the background concepts required for their understanding. The necessary basic definitions from group theory that back these methods can be found in Appendix A.

### 2.1 INVARIANT-EQUIVARIANT AUTOENCODER

Our work builds upon the concept of partial equivariance and the Invariant-Equivariant Autoencoder (IE-AE) shown in Fig. 2 from Winter et al. (2022). Invariant-Equivariant Autoencoders are able to generate a latent representation $(z, g) \in \mathcal{Z} = \{\mathcal{Z}_{\text{inv}}, \mathcal{Z}_{\text{equiv}}\}$ composed of an invariant $z \in \mathcal{Z}_{\text{inv}}$ and an equivariant component $g \in \mathcal{Z}_{\text{equiv}}$. The IE-AE is composed of two main parts: a $\mathcal{G}$-invariant autoencoder, and a $\mathcal{G}$-equivariant group action estimator.

$\mathcal{G}$-**invariant autoencoder.** The first component of the IE-AE is a $\mathcal{G}$-invariant autoencoder $\delta \circ \eta$ composed by a $\mathcal{G}$-invariant encoder $\eta$ and a decoder $\delta$. Its latent space $\mathcal{Z}_{\text{inv}}$ contains $\mathcal{G}$-invariant representations of the input. Note that, because $\eta$ is $\mathcal{G}$-invariant, it holds that:

$$z = \eta(x) = \eta(\rho_{\mathcal{X}}(g)x) \in \mathcal{Z}_{\text{inv}}, \qquad \forall g \in \mathcal{G}, \forall x \in \mathcal{X}. \quad (1)$$

That is, any $\mathcal{G}$-transformation of an input $x \in \mathcal{X}$ yields the same latent representation $z$ in $\mathcal{Z}_{\text{inv}}$. Consequently, the decoder $\delta$ produces identical reconstructions $\hat{x}$ for every transformation the input:

$$\hat{x} = \delta(z) = \delta(\eta(x)) = \delta(\eta(\rho_{\mathcal{X}}(g)x)), \qquad \forall g \in \mathcal{G}, \forall x \in \mathcal{X}. \quad (2)$$

The $\mathcal{G}$-invariant reconstruction $\hat{x}$ corresponds to an element of the orbit of the input $\mathcal{O}_x$ i.e. $\hat{x} = \rho_{\mathcal{X}}(\hat{g}_x)x$ for some $\hat{g}_x \in \mathcal{G}$. This element $\hat{x}$ is denoted as the *canonical representation* of the decoder $\delta$ (or of the input $x$). As indicated by Winter et al. (2022), here "canonical" does not reflect any

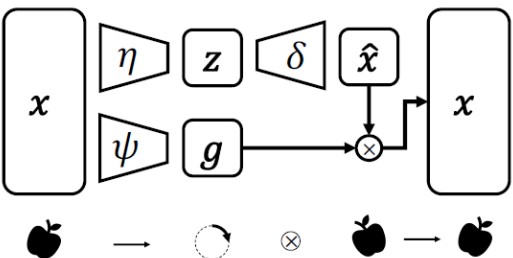

Figure 2: Invariant-Equivariant Autoencoder (IE-AE) architecture (Winter et al., 2022). An IE-AE generates a latent representation $(z, g) \in \mathcal{Z} = \{\mathcal{Z}_{\text{inv}}, \mathcal{Z}_{\text{equiv}}\}$ composed of an invariant $z \in \mathcal{Z}_{\text{inv}}$ and an equivariant component $g \in \mathcal{Z}_{\text{equiv}}$. The invariant part $z$ is obtained through the use of a $\mathcal{G}$-invariant autoencoder $\delta \circ \eta$, and the equivariant part is built through a $\mathcal{G}$-equivariant function $\psi$.

specific property of the element. It simply refers to the orientation $\rho_{\mathcal{X}}(\hat{g}_x)$ learned from the decoder during training, which may depend on various factors and hyperparameters. This is an important observation, as the central mathematical result of our work will concern the convenient collapse of this canonical representation via a constraint in the group action estimator.

**Group action estimator.** The other component of the architecture is the group action estimator $\psi : \mathcal{X} \to \mathcal{G}$. Recall that for a given input $x \in \mathcal{X}$, the canonical representation generated by the $\mathcal{G}$-invariant autoencoder is $\hat{x} = \delta(\eta(x)) = \rho_{\mathcal{X}}(\hat{g}_x)x$, for some group element $\hat{g}_x$. The goal of $\psi$ is to predict the transformation that maps the canonical representation $\delta(\eta(x))$ back to the original $x$. Therefore, such a function $\psi$ must satisfy the property:

$$\rho_{\mathcal{X}}(\psi(x))\,\delta(\eta(x)) = x, \qquad \forall x \in \mathcal{X}. \tag{3}$$

A learnable function satisfying this property is denoted as a *suitable group action estimator*, and it can be constructed as $\psi = \xi \circ \mu$, where $\mu : \mathcal{X} \to \mathcal{Z}_{\text{equiv}}$ is a $\mathcal{G}$-equivariant network, and $\xi : \mathcal{Z}_{\text{equiv}} \to \mathcal{G}$ is a fixed deterministic function that maps the output of $\mu$ to a group element $g \in \mathcal{G}$.

**Training the IE-AE.** Winter et al. (2022) trains all the learnable components of the IE-AE $(\eta, \delta, \psi)$, jointly by optimizing the loss function:

$$\mathcal{L}_1 = d\left(\rho_{\mathcal{X}}(\psi(x))\,\delta(\eta(x)), x\right), \tag{4}$$

where $d$ is a distortion metric, e.g., MSE. Note that Eq. 4 is group invariant by construction. This optimization loss leads to $\mathcal{G}$-invariant representations of the input in the latent space $\mathcal{Z}_{\text{inv}}$, and a $\mathcal{G}$-equivariant estimation $g \in \mathcal{Z}_{\text{equiv}}$ of the transformation needed to reorient $\hat{x}$.

Unlike IE-AEs, our method is not limited to perfect symmetries, and generates consistent, meaningful canonical representations for semantically similar data-points: an ability IE-AEs lacks (Fig. 4).

## 2.2 PARTIAL GROUP EQUIVARIANT CONVOLUTIONAL NEURAL NETWORKS

A map $h : \mathcal{V} \to \mathcal{W}$ is said to be partially equivariant to $\mathcal{G}$ with respect to the representations $\rho_{\mathcal{V}}, \rho_{\mathcal{W}}$ if it is equivariant only to transformations on a *subset* $\mathcal{S}$ of the group $\mathcal{G}$. That is, if $h(\rho_{\mathcal{V}}(g)x) = \rho_{\mathcal{W}}(g)h(x)$, $\forall g \in \mathcal{S} \subseteq \mathcal{G}, \forall x \in \mathcal{X}$ (Romero & Lohit, 2022).[1] Partial G-CNNs can be seen as G-CNNs able to relax their equivariance constraints to hold only on subsets $\mathcal{S} \in \mathcal{G}$ based on the training data. Partial G-CNNs adaptively learn the subsets $\mathcal{S}$ from data by defining a probability distribution on the group from which group elements are sampled during the forward pass at each layer, and learning their parameters during training. In the context of continuous groups, Partial G-CNNs learn connected subsets of group elements $\mathcal{S} = \{\theta^{-1}, ..., e, ..., \theta\} \in \mathcal{G}$ by defining a uniform distribution $\mathcal{U}[-\theta, \theta]$ on the group and learning the value of $\theta$ with the reparameterization trick (Kingma & Welling, 2013; Falorsi et al., 2019). By doing so, Partial G-CNNs are able to fine-tune their equivariance to match the symmetries observed in data, resulting in models with more flexible equivariance constraints than G-CNNs.

It is worth noting that Partial G-CNNs identify partial symmetries at a *dataset level*, whereas our method identifies different symmetry levels for individual dataset elements, without relying on partial convolutions. In addition, our method generalizes beyond uniform distributions. Here, we demonstrate its extension to arbitrary unimodal, symmetric distributions as well as discrete groups.

---

[1]As noted in Romero & Lohit (2022), partial equivariance is in general only approximate because $\mathcal{S}$ is not necessarily closed under $\cdot$. If $\mathcal{S}$ is closed under $\cdot$, then it forms a *subgroup* of $\mathcal{G}$, and $\mathcal{S}$-equivariance is exact.

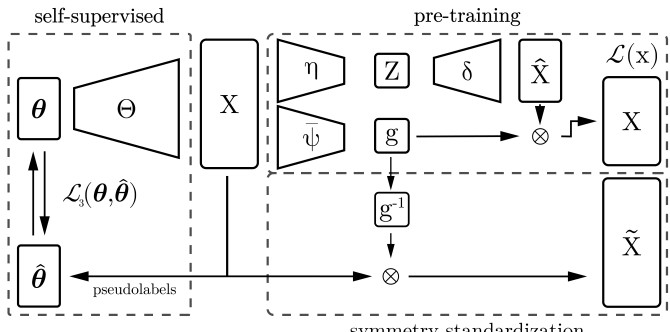

Figure 3: Overview of our proposed method.

# 3 SELF-SUPERVISED DETECTION OF PERFECT AND PARTIAL INPUT-DEPENDENT SYMMETRIES

We aim to learn, without the need for labels, the input-dependent symmetry subsets of the group that accurately represent the symmetries appearing in the data. Similarly to Romero & Lohit (2022), we achieve this by learning a probability distribution $p(u)$ on the group such that $p(u)$ is zero for transformations that do not appear in the data. Focusing on the continuous group of planar rotations $\mathcal{G} = SO(2)$, we consider a uniform distribution $\mathcal{U}[-\theta, \theta]$ defined over a connected set of group elements $\{-\theta, ..., e, ..., \theta\}$. This translates to learning the $\theta$ parameter of the distribution, which we refer to as the *symmetry boundary*. Assuming that datasets can have different symmetry boundaries per sample, this implies that we aim to learn a family of distributions $\mathcal{F} = \{\mathcal{U}[-\theta_x, \theta_x]\}_{x \in \mathcal{X}}$, where the symmetry boundary $\theta_x$ depends on the input $x$.

The proposed method is detailed in Fig. 3. First, we train a modified IE-AE, constrained to encourage meaningful canonical representations across semantically similar inputs (Sec. 3.2), to capture the distribution of symmetries of data. We term this the *pre-training* phase. Next, we use the learned distribution to estimate input-dependent symmetry boundaries, which are then used as pseudo-labels for the self-supervised training of the network $\Theta$. $\Theta$ is ultimately responsible for the prediction of the level of symmetry of the input (Sec. 3.4). We term this the *self-supervision* phase. After training, the IE-AE can be discarded and the $\Theta$ network alone can be used to predict the symmetry level of an input during inference. In addition, we can leverage the inferred canonical representations to transform the original dataset into one in which these symmetries are not present (Sec. 3.6).

## 3.1 LEARNING INPUT-DEPENDING SYMMETRIES FROM DATA

Consider the equivalence relation $\sim_{\mathcal{G}}$ in $\mathcal{X}$ defined by $x \sim_{\mathcal{G}} y$ if and only if $\exists g \in \mathcal{G}$ such that $x = \rho_{\mathcal{X}}(g)y$, and the corresponding quotient set $\mathcal{X}/\sim_{\mathcal{G}}$. Under this definition, two elements of the dataset are related by $\sim_{\mathcal{G}}$ if and only if they are equal up to a $\mathcal{G}$-transformation. Therefore, the equivalence classes $[x] \in \mathcal{X}/\sim_{\mathcal{G}}$ contain the information about the symmetries of each input $x \in \mathcal{X}$.

We assume that the rotation symmetries of every input $x \in \mathcal{X}$ are uniformly distributed in $[-\theta_x, \theta_x]$ for symmetry boundary $\theta_x$ that depends on $x$. We refer to this assumption as *uniformity of symmetries*. Then, every class $[x] \in \mathcal{X}/\sim_{\mathcal{G}}$ has a unique element $c_{[x]} \in [x]$ that is the center of the uniform symmetry, which corresponds to a rotation by zero degrees in the interval $[-\theta_x, \theta_x]$. Note that under the presented framework, the symmetry boundary angle depends on the equivalence class of the input rather than on the input itself, i.e. $\theta_x = \theta_{[x]}$. This is because, as per the definition of $\sim_{\mathcal{G}}$, all elements of a given equivalence class share the same orbit.

With this insight in mind, we can write the elements $s$ of a class $[x]$ as $s = \rho_{\mathcal{X}}(g)c_{[x]}$ for group elements $g$ in $\mathcal{S}_{\theta_{[x]}}$, which is defined as the subset of $G$ with rotation angles in the interval $[-\theta_{[x]}, \theta_{[x]}]$. For instance, in a dataset in which every input has rotational symmetries in $[-60°, 60°]$, the set $\mathcal{S}_{\theta_{[x]}}$ would consist of all the elements of $SO(2)$ with rotation angle in $[-60°, 60°]$. Lastly, let us denote $\psi([x])$ as the images of the elements of $[x]$ obtained by the group action estimator $\psi$. Note that if the rotations in $\psi([x])$ correspond to the rotations in the distribution $\mathcal{U}[-\theta_{[x]}, \theta_{[x]}]$ for each $[x] \in \mathcal{X}/\sim_{\mathcal{G}}$, then $\psi$ is predicting precisely the symmetries appearing in the data. Following the previous example, the equality $\psi([x]) = \mathcal{S}_{\theta_{[x]}}$ would mean that the predictions of $\psi$ for each class are rotations with angles in $[-60°, 60°]$. We are now ready to state the following proposition:

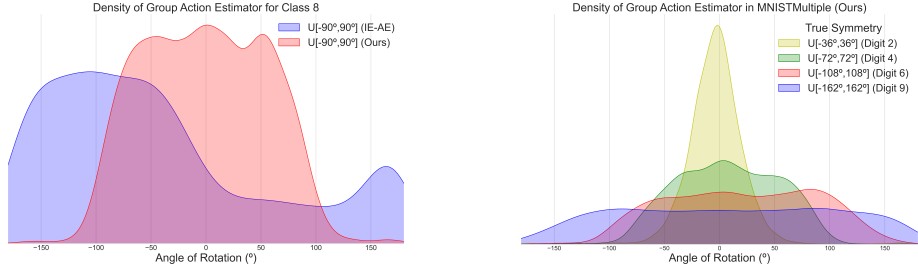

Figure 4: Canonical orientations obtained during inference by the IE-AE (Winter et al., 2022) *(left)* and our method *(right)*. Both models are trained on MNISTRot60-90, a dataset exhibiting uniform rotational symmetries within $[-60°, 60°]$ for digits 0 to 4 and $[-90°, 90°]$ for digits 5 to 9. Our method is able to consistently choose the center of each input's symmetry distribution as the canonical representation (Prop. 3.1 *(ii)*).

Figure 5: Distribution of the transformations predicted by $\psi$ with the IE-AE and our method in class 8 from MNISTRot60-90 *(left)*. Our group action estimator $\bar{\psi}$ correctly captures the different input-dependent distributions the dataset by means of the constraint in Proposition 3.1 *(i) (left)*.

**Proposition 3.1.** *Consider a $\mathcal{G}$-invariant autoencoder $\delta \circ \eta$ and a group action estimator $\psi$. Under the assumption of uniformity of symmetries in $\mathcal{X}$, the following statements are equivalent:*

*(i)* $\psi\left(c_{[x]}\right) = e \; \forall [x] \in \mathcal{X}/{\sim}_G$.

*(ii)* $\forall [x] \in \mathcal{X}/{\sim}_\mathcal{G}$, the canonical representation of any $s \in [x]$ is its center of symmetry $c_{[x]}$.

*(iii)* $\forall [x] \in \mathcal{X}/{\sim}_\mathcal{G}$, it holds that $\psi([x]) = \mathcal{S}_{\theta_{[x]}}$.

*Proof.* See Appendix B. □

The proposition states that, under the constraint introduced in *(i)*, conditions *(ii)* and *(iii)* are satisfied. This implies that a constrained group action estimator $\bar{\psi}$ can effectively collapse the canonical representation of an input into the center of symmetry of its equivalence class, thereby validating its canonical status. Moreover, it ensures that the choice of the canonical representation is consistent across inputs sharing the same invariant representation. This intra-class consistency with meaningful canonical representations is not guaranteed in IE-AEs, as shown in Fig. 4.

Condition *(iii)* states that the symmetries of the class $[x]$, given by $\mathcal{S}_{\theta_{[x]}}$, can be obtained by calculating the image of $[x]$ by $\psi$. This means that, under this constraint, the group action estimator effectively learns the input-dependent distribution of symmetries in the data, unlike in IE-AEs, whose distribution fails to align with the data's inherent symmetries (Fig. 5). In summary, Prop. 3.1 establishes that the constraint in Prop. *(i)* serves as a sufficient and necessary condition to learn subsets of symmetries in the data, and achieve consistent, meaningful canonical representations.

## 3.2 LEARNING CONSTRAINED GROUP ACTION ESTIMATORS

In practice, it is not possible to apply the constraint $\psi(c_{[x]}) = e \; \forall [x] \in \mathcal{X}/{\sim}_\mathcal{G}$ to the group action estimator, as we do not know a priori which elements in the dataset are centers of symmetry. However, we can encourage convergence to this solution by minimizing $d(\psi(x), e)$, with $d$ a distorsion metric, e.g., MSE (see Appendix B.1). Combining this term with the loss term of Eq. 4, we obtain an optimization loss:

$$\mathcal{L} = \mathcal{L}_1 + \mathcal{L}_2 = d_1\left(\rho_\mathcal{X}(\psi(x))\, \delta(\eta(x)), x\right) + d_2\left(\psi(x), e\right). \quad (5)$$

Jointly optimizing $\mathcal{L}_1$ and $\mathcal{L}_2$ encourages convergence to solutions that comply to Proposition 3.1.

### 3.3 Estimation of the symmetry boundary

Suppose we have pretrained our constrained group action estimator using the loss $\mathcal{L}$ as per Eq. 5. Prop. 3.1 *(iii)* states that the symmetry distribution of an input $x$ is defined by the image of its equivalence class $[x]$ under $\psi$. However, calculating these equivalence classes is infeasible in practice, since $SO(2)$ is an infinite group. To overcome this problem, we base our approach on the convenient assumption that *similar objects share the same distribution of symmetries*. This assumption is grounded in empirical observations across diverse real-world scenarios in which different objects within the same category exhibit consistent symmetrical patterns, such as a shared rotational distribution. In essence, this implies that inputs that are semantically similar to $x$ contain information about its symmetries, much like the equivalence class $[x]$.

Motivated by this observation, we shift from using equivalence classes $[x]$ to sets of objects semantically similar to $x$ for estimating $\theta_x$. This approach bypasses the need for calculating equivalence classes, and allows for a self-supervised estimation of $\theta_x$ through the use of pseudo-labels.

### 3.4 Self-Supervised learning of the symmetry boundaries

**Generating the pseudo-labels.** Let $\mathcal{N}_{k,d} = \mathcal{N}_k : \mathcal{X} \to \mathscr{P}(\mathcal{X})$ be a function that maps each input $x$ to the set $\mathcal{N}_{k,d}(x) \subset \mathcal{X}$ of $k$-neighbors around $x$ in the $\mathcal{G}$-invariant latent space $\mathscr{Z}_{\text{inv}}$ as measured by some distance metric $d$, i.e., the $k$ elements of the dataset whose $\mathcal{G}$-invariant embeddings are closest to the $\mathcal{G}$-invariant embedding of $x$, $\eta(x)$. $\mathcal{N}_k(x)$ acts as a substitute of the equivalence class $[x]$, and contains elements semantically similar to $x$, which we assume to share the symmetry distribution of $x$. We then estimate the level of symmetry of $x$ within the dataset $\mathcal{X}$ by estimating the parameter of the distribution corresponding to $\psi(\mathcal{N}_k(x))$ with an estimator $E$ of uniform distributions of the form $\mathcal{U}[-\theta, \theta]$. In practice, we found beneficial to convert the original distribution $\mathcal{U}[-\theta, \theta]$ to a distribution $\mathcal{U}[0, \theta]$ by taking the absolute value of $\psi$'s predictions. We then use the Method of Moments estimator for uniform distributions of this form, which proved to be more robust to outliers than other estimators, resulting in more reliable pseudo-labels (Appx. C). Combining all components, we calculate the pseudo-labels for the estimation of the symmetry boundary $\theta_x$ as:

$$\hat{\theta}_x = \left( E \circ |\cdot| \circ \psi \circ \mathcal{N}_k \right)(x) \tag{6}$$

**Learning the levels of symmetry.** Once we calculate the pseudo-labels, we can use them to learn the levels of symmetry of each input $x \in \mathcal{X}$ in a self-supervised manner. To this end, we introduce a *boundary prediction network* $\Theta = \omega \circ \phi : \mathcal{X} \to \mathbb{R}^+$ consisting of a $\mathcal{G}$-invariant network $\phi$, followed by a fully connected network $\omega$. The boundary prediction network $\Theta$ is trained to minimize the difference between the predicted symmetry boundary $\Theta(x)$, and the estimated pseudo-label $\hat{\theta}_x$:

$$\mathcal{L}_3 = d\left( \Theta(x), E \circ |\cdot| \circ \psi \circ \mathcal{N}_k(x) \right) = d\left( \Theta(x), \hat{\theta}_x \right). \tag{7}$$

The $\mathcal{G}$-invariance in $\Theta$ reflects that all samples on an orbit share the same level of symmetry $\theta_{[x]}$.

### 3.5 Symmetry standardization

Our method also allows for the removal of symmetries in a given dataset based on its symmetric properties. Prop. 3.1 states that, under the constraint outlined in Prop. 3.1 *(i)*, the canonical representation of every element $x \in [x]$ is the center of symmetry $c_{[x]}$ for all $[x] \in \mathcal{X}/\sim_G$. Let $\overline{\psi}$ denote a group action estimator subject to this constraint. Recall that any $x \in \mathcal{X}$ belongs to an equivalence class $[x]$ with a unique center of symmetry $c_{[x]}$. Since $\overline{\psi}$ is suitable, it holds that:

$$x = \rho_{\mathcal{X}}(\overline{\psi}(x))\delta(\eta(x)) = \overset{\overset{\text{Prop. 3.1 (ii)}}{\uparrow}}{\rho_{\mathcal{X}}(\overline{\psi}(x))\hat{x}} = \rho_{\mathcal{X}}(\overline{\psi}(x))c_{[x]} \iff \rho_{\mathcal{X}}(\overline{\psi}(x)^{-1})x = c_{[x]}. \tag{8}$$

That is, the inverse of the group actions predicted by $\overline{\psi}$ can be used to reorient the input towards the center of symmetry of its class. This can be done efficiently for every input without involving the calculation of the classes $[x]$. Then, the set $\tilde{\mathcal{X}} = \left\{ \rho_{\mathcal{X}}(\overline{\psi}(x)^{-1})x \right\}_{x \in \mathcal{X}}$, is a standardized, $\mathcal{G}$-invariant version of the data $\mathcal{X}$ whose symmetries have been effectively removed by collapsing every input into the orientation of the center of symmetry of its class.

### 3.6 Considering other symmetry distributions

Our method is not restricted to uniform symmetry distributions. It can consider other symmetry distributions –both continuous and discrete– by adjusting the hypotheses in Prop. 3.1, and deriving

appropriate estimators for the pseudo-labels. Furthermore, we show that, for arbitrary unimodal, symmetric distributions, the objective $\mathcal{L}_2$ converges to the center of symmetry $c_{[x]}$, allowing us to learn and predict symmetry levels through the previous construction (Appx. B.1).

**Gaussian symmetries.** Consider a dataset $\mathcal{X}$, whose elements are governed by rotational symmetries $\mathcal{S}_{\sigma_{[x]}} \subset \mathcal{G}$ sampled from a Gaussian distributions $N(0, \sigma_{[x]})$, whose center is defined as the center of symmetry $c_{[x]}$. Under these assumptions, the following proposition holds:

**Proposition 3.2.** *Consider a $\mathcal{G}$-invariant autoencoder $\delta \circ \eta$ and a group action estimator $\psi$. Under Gaussian symmetries in $\mathcal{X}$, the following statements are equivalent:*

*(i)* $\psi\left(c_{[x]}\right) = e \; \forall [x] \in \mathcal{X}/\sim_G$.

*(ii)* $\forall [x] \in \mathcal{X}/\sim_{\mathcal{G}}$, the canonical representation of any $s \in [x]$ is its center of symmetry $c_{[x]}$.

*(iii)* $\forall [x] \in \mathcal{X}/\sim_{\mathcal{G}}$, it holds that $\psi([x]) = \mathcal{S}_{\sigma_{[x]}}$.

*Proof.* See Appendix B. ☐

**Discrete symmetries.** We can also consider datasets governed by discrete symmetric groups, e.g., the cyclic order $n$ subgroups of $\mathrm{SO}(2)$, $\mathcal{C}_n$. In this case, each dataset sample $x$ shows cyclic symmetries given by a group $\mathcal{S}_{n_{[x]}} = \mathcal{C}_{n_{[x]}}$ of order $n_{[x]} \in \mathbb{N}$. Importantly, note that since cyclic groups are intrinsically symmetric, *every element of the group is equally valid to serve as center of symmetry*. This property of symmetric discrete groups lets us relax the condition $(i)$ in Prop. 3.3:

**Proposition 3.3.** *Consider a $\mathcal{G}$-invariant autoencoder $\delta \circ \eta$ and a group action estimator $\psi$. Under cyclic symmetries in $\mathcal{X}$, the following statements are equivalent:*

*(i)* $\psi\left(c_{[x]}\right) = e \; \forall [x] \in \mathcal{X}/\sim_G$ for some $c_{[x]} \in [x]$.

*(ii)* $\forall [x] \in \mathcal{X}/\sim_{\mathcal{G}}$, the canonical representation of any $s \in [x]$ is the element $c_{[x]} \in [x]$.

*(iii)* $\forall [x] \in \mathcal{X}/\sim_{\mathcal{G}}$, it holds that $\psi([x]) = C_{n_{[x]}}$.

*Proof.* See Appendix B. ☐

As shown in Appx. C, we can construct proper group action estimators for each of these symmetry distributions in order to generate appropriate pseudo-labels for $\Theta$.

## 4 RELATED WORK

**Unsupervised learning of invariant and equivariant representations.** Unsupervised learning of both invariant and equivariant representations through the use of autoencoder-based approaches has been previously proposed (Winter et al., 2022; Shu et al., 2018; Winter et al., 2021; Yokota & Hontani, 2022; Guo et al., 2019; Feige, 2019; Kosiorek et al., 2019; Koneripalli et al., 2020). However, existing methods, e.g., Quotient Autoencoders (QAE) (Yokota & Hontani, 2022), Invariant-Equivariant Autoencoders (Winter et al., 2022) obtain arbitrary preferred orientations –or canonical representation. In contrast to existing approaches, our proposed work is not limited to perfect symmetries, and is able to learn meaningful consistent canonical representations.

**Soft equivariance and soft invariance.** Typical group equivariant approaches do not inherently learn their level of symmetry based on data. Instead, these symmetries are imposed manually through the choice of the group prior to training (Cohen & Welling, 2016; Cohen et al., 2018; Weiler et al., 2018; Weiler & Cesa, 2019; Cohen et al., 2019; Romero et al., 2020; Romero & Cordonnier, 2020; Wang et al., 2020). This approach has limitations when dealing with datasets that contain partial symmetries –such as real-world images– resulting in overly constrained models. In such cases, *soft* or *partial* equivariance (or invariance) is desired, allowing these properties to hold only for a subset of the group transformations. Canonical examples are Augerino (Benton et al., 2020) and Partial G-CNNs (Romero & Lohit, 2022), which achieve this by learning a probability distribution over transformations. Other approaches handle soft equivariance through a combination of equivariant and non-equivariant models (Finzi et al., 2021a;b). Nevertheless, existing approaches generally require supervised training, and only capture levels of symmetry at a dataset-level. In contrast, our method is able to learn levels of symmetry at a sample-level, and does so in a self-supervised manner.

**Symmetry standardization.** Finally, Spatial Transformer Networks (STN) Jaderberg et al. (2015) transform the input to counteract data transformations through a learnable projective operation. This

is akin to our data standardization process. However, STNs are usually trained in a supervised manner, as part of a broader network for tasks like classification. Instead, we produce symmetry standardization without relying on labelled examples.

## 5 EXPERIMENTS

In this section, we evaluate our approach. Comprehensive implementation details, including architecture specifications and optimization techniques, can be found in Appx. D.

**Prediction of input-dependent levels of symmetry.** To evaluate the ability of our method to predict the levels of symmetry, we use standard and synthetic versions of the MNIST-12ᴋ (LeCun et al., 1998) dataset, divided into $12,000$ train and $50,000$ test images. We construct RotMNIST60, an MNIST variation with digits uniformly rotated in the interval $[-60°, 60°]$; RotMNIST60-90, a variation with digits uniformly rotated in $[-60°, 60°]$ for the classes $0-4$, and in $[-90°, 90°]$ for the classes $5-9$; and MNISTMultiple, a MNIST variation with different rotational symmetries per class starting from zero rotation for the class 0, and increasing the maximum rotation by $18°$ in each class, i.e., $[-18°, 18°]$ for class 1, $[-36°, 36°]$ for class 2, etc. We also create a Gaussian variation of MNISTMultiple, MNISTGaussian, which exhibits rotational Gaussian distributions with increasing standard deviations per-class, following multiples of $9°$. This choice ensures that $95\%$ of sampled rotations falls within the corresponding MNISTMultiple's interval. For the cyclic case, we construct MNISTC2-C4, an analogous to MNISTRot60-90 where rotations are drawn from $\mathcal{C}_2$ and $\mathcal{C}_4$ for the corresponding class subsets. We additionally evaluate our method on the standard MNIST and rotated MNIST (RotMNIST) dataset (Larochelle et al., 2007). We consider two metrics in our evaluation: the average predicted symmetry level $\overline{\Theta}$, and the Mean Absolute Error (MAE) between the predictions and the true boundary angles $\theta$. All metrics are calculated in the test set, and the best model is chosen based on best loss obtained during validation. Our results are summarized in Fig. 6 and shown in an extended format in Tabs. 3, 4. For datasets with full symmetries (MNISTRot) and no symmetries (MNIST), our method obtains consistently accurate predictions of the symmetry levels across all classes. For MNIST, we note minor deviations from the expected symmetry level of $0°$ across all classes. Rather than an imprecision, we attribute this to the inherent rotational symmetries proper of handwritten digits caused by diverse writing styles and nuances.

In datasets with partial symmetries, our model consistently identifies the correct level of symmetry across all classes. This is true even when various levels of symmetry are present within a single dataset. For instance, the MNISTRot60 experiment highlights our model's ability to adapt to partial symmetries, while results from the MNISTRot60-90 experiment indicate its capability to discern varying levels of symmetry on a per-class basis. In the challenging MNISTMultiple dataset, our model consistently predicts varying symmetry levels for each class, showcasing its ability to handle diverse, intricate symmetry scenarios. Furthermore, results on MNISTGaussian show that our method generalizes to other unimodal, symmetric distributions, even in challenging scenarios.

On MNISTC2-C4 we observe that our model presents some inaccuracies for the classes 3, 4, and 5. However, looking at the per-class density estimations (Fig. 7, Appx. D) confirms that $\psi$ accurately captures data symmetries as outlined in Prop.3.3. The observed imprecision stems likely from the limitations of our density-comparison approach for generating pseudo-labels, which is error-prone due to requiring very high number of neighbours per-input. Implementing more sophisticated methods to estimate cyclic groups from densities could enhance the precision of these estimations.

**Out-of-distribution symmetry detection.** We further validate out method for the detection of objects whose symmetries differ from those observed during training. To this end, fully rotated digits are passed through a classifier, whose task is to predict whether object symmetries have been seen during training or not. These out-of-distribution classifiers use the models trained on the previous section, and are tested on fully-rotated unseen inputs without further training (see Appx. D.2).

As shown in Table 1, all models consistently demonstrate their ability to identify unseen symmetries during inference. As expected, there is a modest decline in the performance of the out-of-distribution detection models as the complexity of the symmetries in the data increases. Nevertheless, even in dataset with intricate and multiple symmetries, we achieve accuracies of $86.48\%$ and $83.64\%$, which underscores the potential for practical applications of our method.

Table 1: Test accuracy results for out-of-distribution symmetry detection.

|  | ACCURACY |
|---|---|
| ROTMNIST | 92.27% |
| ROTMNIST60-90 | 91.30% |
| MNISTMULTIPLE | 86.48% |
| MNISTGAUSSIAN | 83.64% |
| MNISTC2-C4 | 82.47% |

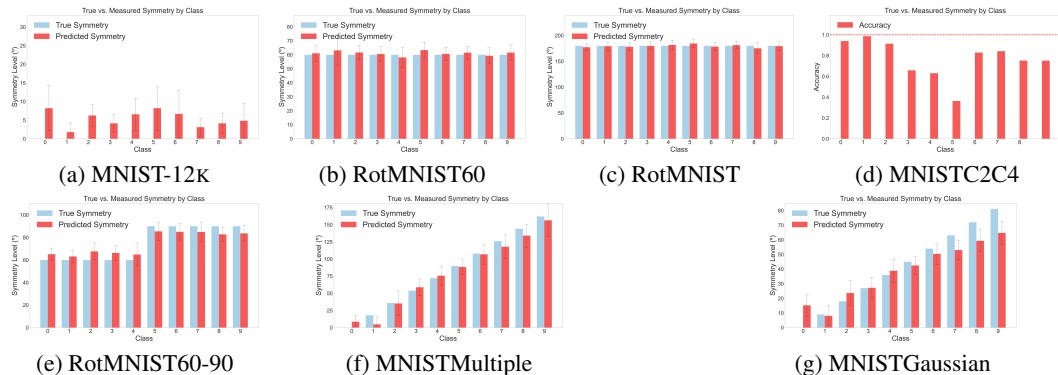

Figure 6: Prediction of input-dependent levels of symmetry on synthetic MNIST variants.

Table 2: Comparison of test set accuracy scores of baseline supervised and unsupervised models (ResNet-18 and K-Means respectively), with and without the use of symmetry standardization.

| Dataset | RESNET-18 | | K-MEANS | | IE-AE + KNN |
|---|---|---|---|---|---|
| | Regular | Symmetry Std. | Regular | Symmetry Std. | |
| MNISTROT60 | **97.47**% | 97.39% | 65.585% | **86.81**% | 95.58% |
| MNISTROT60-90 | 96.99% | **97.23**% | 64.74% | **86.36**% | 95.31% |
| MNISTMULTIPLE | 96.79% | **97.16**% | 61.93% | **86.22**% | 95.80% |
| MNISTGAUSSIAN | **96.70**% | 96.58% | 65.00% | **86.06**% | 95.65% |
| MNISTC2C4 | 97.23% | **97.53**% | 66.01% | **81.83**% | 95.22% |
| MNISTROT | 95.39% | **96.35**% | 42.70% | **83.61**% | 95.25% |

**Improving non-equivariant models with symmetry standardization.** To conclude, we investigate the impact of symmetry standardization in the performance of non-equivariant models. To achieve this, we consider baseline supervised and unsupervised learning models: ResNet-18 (He et al., 2015) and K-Means, and compare the performances obtained with the original datasets against those obtained using the symmetry-standardized versions generated by our method. As shown in Tab. 2, both unsupervised and supervised methods see performance enhancements from symmetry standardization, with a particularly pronounced effect for K-Means.

To further evaluate the advantages of symmetry standardization, we compare it with a supervised K-Nearest Neighbors (KNN) classifier trained on the $\mathcal{G}$-invariant embeddings of an IE-AE (Tab. 2). While IE-AE embeddings offer improvements thanks to their $\mathcal{G}$-invariance, they can only be used as embeddings for other models. In contrast, symmetry standardization directly transfers the $\mathcal{G}$-invariance to the input data itself. allowing subsequent methods to operate on the data itself without being constrained by the lower dimensional embeddings of an IE-AE. It is worh noting that, in principle, symmetry standardization could also be achieved with regular IE-AEs. However, it is essential to have *consistent, meaningful canonical representations to collapse to during this process* –a property that IE-AEs lack (Fig. 4). We achieve this through the constraint proposed in Prop. 3.1*(i)*.

## 6  CONCLUSIONS AND LIMITATIONS

We proposed a method to determine the distribution of symmetries for each input in the dataset in a self-supervised manner. We showed through various experiments the effectiveness of our method, showcasing its ability to adapt to multiple, complex symmetries –both perfect and partial– within a single dataset. Furthermore, our method is able to accommodate different families of symmetry distributions and offers practical benefits, notably in out-of-distribution symmetry detection.

**Limitations.** The main limitation of our method is related to the symmetry distributions it is able to represent. Specifically, the main assumption for the finding of $c_{[x]}$ is that the underlying symmetry distribution is both unimodal and symmetric. Extending our results to complex, multimodal symmetry distributions is an interesting direction for future research. In addition, our method is limited by the group actions that can be represented by an IE-AE. We observe that in datasets with high intra-class variability, e.g., CIFAR10, objects within the same class may not share a $\mathcal{G}$-invariant representation that connects them through a group action. In such cases, the standard MSE loss used in IE-AE is unable to capture this relationship (Eq. 4), limiting our method. In the future, we aim to mitigate this through the use of more semantically meaningful metrics, e.g., SSIM (Wang et al., 2004).

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

# APPENDIX

## A  BACKGROUND

**Groups and group actions.** A group $\mathcal{G}$ is a set equipped with a closed, associative binary operation $\cdot$ such that $\mathcal{G}$ contains an identity element $e \in G$ and every element $g \in \mathcal{G}$ has an inverse $g^{-1} \in \mathcal{G}$. For a given set $\mathcal{X}$ and group $\mathcal{G}$, the (left) group action of $\mathcal{G}$ on $\mathcal{X}$ is a map $\rho : \mathcal{G} \times \mathcal{X} \to \mathcal{X}$ that preserves the group structure. Intuitively, it describes how set elements transform by group elements.

**Group representations.** In this work, we focus on cases where $\mathcal{X}$ is a vector space. In such scenarios, the group acts on it by means of *group representations*. Specifically, a representation of $\mathcal{G}$ is a function $\rho_{\mathcal{X}} : \mathcal{G} \to \mathrm{GL}(\mathcal{X})$ that maps each group element to an invertible n×n matrix from the *general linear group* $\mathrm{GL}(X)$, where n is the dimension of the vector space $\mathcal{X}$. We consider our datasets to be of the form $\mathcal{X} = \{f \,|\, f : \mathcal{V} \to \mathcal{W}\}$ where $\mathcal{V}$ and $\mathcal{W}$ are vector spaces. For instance, an RGB image can be interpreted as a function $f : \mathbb{R}^2 \to \mathbb{R}^3$ that maps each pixel location to a three-channel intensity value. Following this definition, a group element acts in a data sample as:

$$[\rho_{\mathcal{X}}(g)f](x) \equiv \rho_{\mathcal{W}}(g)f\left(\rho_{\mathcal{V}}(g^{-1})x\right). \tag{9}$$

For the reminder of this paper, when we refer to representations $\rho_{\mathcal{X}}$ on $\mathcal{X}$, it is understood that we are implicitly referring to the previous equation to understand the transformation of each component.

**Orbits.** A central concept in our study is the orbit of $x$, defined as $\mathcal{O}_x = \{\rho_{\mathcal{X}}(g)x\}_{g \in \mathcal{G}}$. The orbit of $x$ captures all possible transformations of $x$ resulting from the action of all elements of $\mathcal{G}$.

**Equivalence classes and quotient sets.** Our analysis strongly relies on the definition of equivalence classes and their quotient sets. Let ~ be an equivalence relation on $\mathcal{X}$ and consider the *equivalence classes* $[x] = \{y \in \mathcal{X}, \text{ s.t. } x \sim y\}$ of $\mathcal{X}$. The quotient set $\mathcal{X}/\sim$ is defined as the collection of all equivalent classes in $\mathcal{X}$ under the relation ~.

**Group equivariance and group invariance.** A map $h : \mathcal{V} \to \mathcal{W}$ is $\mathcal{G}$-equivariant with respect to the representations $\rho_V, \rho_W$ if $h(\rho_V(g)x) = \rho_{\mathcal{W}}(g)h(x) \; \forall g \in \mathcal{G}, \forall x \in \mathcal{X}$. In the context of neural networks, G-CNNs (Cohen & Welling, 2016) are designed to be $\mathcal{G}$-equivariant by using only $\mathcal{G}$-equivariant layers in their constructions. This ensures that applying a transformation $g \in \mathcal{G}$ before or after a layer yields the same result. Analogously, a map $h$ is $\mathcal{G}$-invariant with respect to $\rho_V$ if $h(\rho_V(g)x) = h(x) \; \forall g \in \mathcal{G}, \forall x \in \mathcal{X}$. That is, if $\mathcal{G}$-transformations of the input yield the same result.

## B  PROOFS

**Proposition B.1.** *Consider a $\mathcal{G}$-invariant autoencoder $\delta \circ \eta$ and a group action estimator $\psi$. Under the assumption of uniformity of symmetries in $\mathcal{X}$, the following statements are equivalent:*

*(i)* $\psi\left(c_{[x]}\right) = e \; \forall [x] \in \mathcal{X}/\sim_G$.

*(ii)* $\forall [x] \in \mathcal{X}/\sim_{\mathcal{G}}$, *the canonical representation of any $s \in [x]$ is its center of symmetry $c_{[x]}$.*

*(iii)* $\forall [x] \in \mathcal{X}/\sim_{\mathcal{G}}$, *it holds that $\psi([x]) = \mathcal{S}_{\theta_{[x]}}$.*

*Proof.* Let us prove the proposition by proving *(i)* $\Longleftrightarrow$ *(ii)* and *(i)* $\Longleftrightarrow$ *(iii)*.

*(i)* $\Longrightarrow$ *(ii)* Let $[x] \in \mathcal{X}/\sim_{\mathcal{G}}$. Because of uniformity of symmetries, we can write the elements of $[x]$ as $s = \rho_{\mathcal{X}}(g)c_{[x]} \in [x]$, where $g \in \mathcal{S}_{\theta_{[x]}}$. We want to prove that its canonical representation is the center of symmetry of the class, $c_{[x]}$. The canonical representation of $s$ is given by

$$\hat{s} = \delta(\eta(s)) = \delta(\eta(\rho_{\mathcal{X}}(g)c_{[x]})) \overset{\overset{\eta \;\; G-inv}{\uparrow}}{=} \delta(\eta(c_{[x]})) \tag{10}$$

Now let us calculate the canonical representation of the center of symmetry. Because $\psi$ is suitable, we know that

$$\rho_{\mathcal{X}}(\psi(x))\,\delta(\eta(x)) = x \quad \forall x \in X \tag{11}$$

In particular,

$$\rho_X(\psi(c_{[x]}))\,\delta(\eta(c_{[x]})) = c_{[x]} \tag{12}$$

Expanding the left-hand side of the previous equation, we get

$$\rho_{\mathcal{X}}(\underbrace{\psi(c_{[x]})}_{\psi(c_{[x]})=e \ \forall c_{[x]} \in \mathcal{X}/\sim_{\mathcal{G}}})\delta(\eta(c_{[x]})) = \delta(\eta(c_{[x]})) \tag{13}$$

and joining equation 10, equation 12 and equation 13 we get

$$\hat{s} = \delta(\eta(c_{[x]})) = c_{[x]} \tag{14}$$

as we wanted to show.

*(ii)* $\Longrightarrow$ *(i)* Let us assume that $\hat{s} = \delta(\eta(s)) = c_{[x]} \ \forall s \in [x] \ \forall [x] \in \mathcal{X}/\sim_{\mathcal{G}}$. Let $s \in [x]$. Because $\psi$ is suitable, it holds that

$$s = \rho_{\mathcal{X}}(\psi(s))\underbrace{\delta(\eta(s))}_{\delta(\eta(s))=c_{[x]}} = \rho_{\mathcal{X}}(\psi(s))c_{[x]} \tag{15}$$

In particular, for $s = c_{[x]}$

$$c_{[x]} = \rho_{\mathcal{X}}(\psi(c_{[x]}))c_{[x]} \iff \psi(c_{[x]}) = e \tag{16}$$

as we wanted to prove.

*(i)* $\Longrightarrow$ *(iii)* Suppose that $\psi(c_{[x]}) = e \ \forall [x] \in \mathcal{X}/\sim_{\mathcal{G}}$. Let us prove that the group function $\psi$ is predicting exactly the symmetries in the data, which are given by $\mathcal{S}_{\theta_{[x]}}$. Let $[x] \in X/\sim_G$. Because of uniformity of symmetries, we can write the elements of $[x]$ as $s = \rho_{\mathcal{X}}(g)c_{[x]} \in [x]$, where $g \in \mathcal{S}_{\theta_{[x]}}$. Therefore,

$$\psi(s) = \underbrace{\psi(\rho_{\mathcal{X}}(g)c_{[x]})}_{\psi \ G-equiv} = g \cdot \underbrace{\psi(c_{[x]})}_{\psi(c_{[x]})=e \ \forall c_{[x]} \in \mathcal{X}/\sim_{\mathcal{G}}} = g \in \mathcal{S}_{\theta_{[x]}} \tag{17}$$

This is, the transformations predicted by $\psi$ on $[x]$ are the elements of $\mathcal{S}_{\theta_{[x]}}$ i.e. the symmetry distribution of $[x]$ (and *vice versa*), as we wanted to prove.

*(iii)* $\Longrightarrow$ *(i)* Suppose that the transformations in $\psi([x])$ are those in the data, $\mathcal{S}_{\theta_{[x]}}$. Let us prove by contradiction that $\psi(c_{[x]}) = e \ \forall [x] \in \mathcal{X}/\sim_{\mathcal{G}}$. Therefore, suppose that exists a $[x]_0 \in \mathcal{X}/\sim_G \ s.t. \ \psi(c_{[x]_0}) \neq e$. Let $\mathcal{S}_{\theta_{[x]_0}}$ its symmetry distribution in the data.

*Case 1.* $\psi(c_{[x]_0}) = g_0 \notin \mathcal{S}_{\theta_{[x]_0}}$

It is clear that $\psi(c_{[x]_0})$ can not take the value of an element outside of the subset $\mathcal{S}_{\theta_{[x]_0}}$, as then we would have found a class $[x]_0$ with an element $c_{[x]_0} \in [x]_0$ whose image by $\psi$ is not a transformation with angle in $[-\theta_0, \theta_0]$, which is a contradiction with $\psi([x]) = \mathcal{S}_{\theta_{[x]}}$ for all $[x] \in \mathcal{X}/\sim_{\mathcal{G}}$.

*Case 2.* $\psi(c_{[x]_0}) = g_0 \in \mathcal{S}_{\theta_{[x]_0}}, \ \ g_0 \neq e$

Consider an element of the class $s \in [x]_0$. Because of uniformity of symmetries, we can write the elements of $[x]_0$ as $s = \rho_{\mathcal{X}}(g)c_{[x]_0} \in [x]_0$, where $g \in \mathcal{S}_{\theta_{[x]_0}}$. Then,

$$\psi(s) = \underbrace{\psi(\rho_{\mathcal{X}}(g)c_{[x]_0})}_{\psi \ G-equiv} = g \cdot \underbrace{\psi(c_{[x]_0})}_{\psi(c_{[x]_0})=g_0 \in \mathcal{S}_{\theta_{[x]_0}}} = g \cdot g_0 \tag{18}$$

Now consider the elements $s_l, s_u \in [x]_0$ that are at the lower and upper bound respectively of the uniform symmetry of $[x]_0$. Then, $s_l = \rho_{\mathcal{X}}(g_{-\theta_0})c_{[x]_0}$ and $s_u = \rho_{\mathcal{X}}(g_{\theta_0})c_{[x]_0}$ where $g_{-\theta_0}, g_{\theta_0} \in \mathcal{S}_{\theta_{[x]_0}}$ are the transformations whose rotation angles are $-\theta_0$ and $\theta_0$ respectively. Consider their

images by $\psi$ as given by equation 18

$$\psi(s_l) = g_{-\theta_0} \cdot g_0, \quad \psi(s_u) = g_{\theta_0} \cdot g_0 \tag{19}$$

Because $g_0 \neq e$, then $g_0 = g_\alpha$ for some angle $\alpha$ that is strictly positive or strictly negative. If $\alpha > 0$, then $\psi(s_u) = g_{\theta_0} \cdot g_\alpha = g_{\theta_0 + \alpha}$ which is not in $\mathcal{S}_{\theta_{[x]_0}}$. Similarly, if $\alpha < 0$ then $\psi(s_l) = g_{-\theta_0 + \alpha} \notin \mathcal{S}_{\theta_{[x]_0}}$. In any case, we found an element of $[x]_0$ whose image by $\psi$ is a transformation with rotation angle not in $[-\theta_0, \theta_0]$, which is a contradiction with $\psi([x]) = \mathcal{S}_{\theta_{[x]}}$ for all $[x] \in \mathcal{X}/\sim_{\mathcal{G}}$.

Therefore, $\psi(c_{[x]}) = e \; \forall [x] \in X/\sim_G$ as we wanted to show. $\qquad \square$

**Proposition B.2.** *Consider a $\mathcal{G}$-invariant autoencoder $\delta \circ \eta$ and a group action estimator $\psi$. Under Gaussian symmetries in $\mathcal{X}$, the following statements are equivalent:*

*(i)* $\psi\left(c_{[x]}\right) = e \; \forall [x] \in \mathcal{X}/\sim_G$.

*(ii)* $\forall [x] \in \mathcal{X}/\sim_{\mathcal{G}}$, *the canonical representation of any $s \in [x]$ is its center of symmetry $c_{[x]}$.*

*(iii)* $\forall [x] \in \mathcal{X}/\sim_{\mathcal{G}}$, *it holds that $\psi([x]) = \mathcal{S}_{\sigma_{[x]}}$.*

**Remark B.1.** *Note that because the angles in $\mathcal{S}_{\sigma_{[x]}}$ are sampled from a Gaussian distribution $N(0, \sigma_{[x]})$, the set $\mathcal{S}_{\sigma_{[x]}}$ is not guaranteed to contain the identity element, corresponding to exactly $0°$ in the Gaussian distribution sample $S \sim N(0, \sigma_{[x]})$. For the sake of simplicity, we assume $e \in \mathcal{S}_{\sigma_{[x]}}$. In practice, this assumption does not present a problem, as the objective $\mathcal{L}_2$ ensures convergence of the canonical representation to the center of the Gaussian distribution (see Appendix B.1). Additionally, we also assume that the samples from the $N(0, \sigma_{[x]})$ are not (by chance) degenerate ($\mathcal{S}_{\sigma_{[x]}} = \{e\}$), or cyclic ($\mathcal{S}_{\sigma_{[x]}} = C_n$), case whose proof is covered in Prop.(ref).*

*Proof.* Let us prove the proposition by proving *(i)* $\Longleftrightarrow$ *(ii)* and *(i)* $\Longleftrightarrow$ *(iii)*.

*(i)* $\Longrightarrow$ *(ii)*, *(ii)* $\Longrightarrow$ *(i)*, *(i)* $\Longrightarrow$ *(iii)* Same as Prop. 3.1 but for $\theta_{[x]} = \sigma_{[x]}$ and substituting the uniformity of symmetries condition by Gaussian symmetries condition.

*(iii)* $\Longrightarrow$ *(i)* Suppose that the transformations in $\psi([x])$ are those in the data, $\mathcal{S}_{\sigma_{[x]}}$. Let us prove by contradiction that $\psi(c_{[x]}) = e \; \forall [x] \in \mathcal{X}/\sim_{\mathcal{G}}$. Therefore, suppose that exists a $[x]_0 \in \mathcal{X}/\sim_G$ *s.t.* $\psi(c_{[x]_0}) \neq e$. Let $\mathcal{S}_{\sigma_{[x]_0}}$ its symmetry distribution in the data.

*Case 1.* $\psi(c_{[x]_0}) = g_0 \notin \mathcal{S}_{\sigma_{[x]_0}}$

It is clear that $\psi(c_{[x]_0})$ can not take the value of an element outside of the subset $\mathcal{S}_{\sigma_{[x]_0}}$, as then we would have found a class $[x]_0$ with an element $c_{[x]_0} \in [x]_0$ whose image by $\psi$ is not a transformation with angle in $S \sim N(0, \sigma_{[x]_0})$, which is a contradiction with $\psi([x]) = \mathcal{S}_{\sigma_{[x]}}$ for all $[x] \in \mathcal{X}/\sim_{\mathcal{G}}$.

*Case 2.* $\psi(c_{[x]_0}) = g_0 \in \mathcal{S}_{\sigma_{[x]_0}}, \quad g_0 \neq e$

Consider an element of the class $s \in [x]_0$. Under Gaussian symmetries, we can write the elements of $[x]_0$ as $s = \rho_{\mathcal{X}}(g)c_{[x]_0} \in [x]_0$, where $g \in \mathcal{S}_{\sigma_{[x]_0}}$. Then,

$$\psi(s) = \psi(\rho_{\mathcal{X}}(g)c_{[x]_0}) \overset{\substack{\psi \; G\text{-}equiv \\ \uparrow}}{=} g \cdot \psi(c_{[x]_0}) \underset{\substack{\downarrow \\ \psi(c_{[x]_0}) = g_0 \in \mathcal{S}_{\sigma_{[x]_0}}}}{=} g \cdot g_0 \tag{20}$$

Proceeding as in Prop. 3.1, if we find an element $s \in [x]_0$ such that $\psi(s) \notin \mathcal{S}_{\sigma_{[x]_0}}$, we will have proved the result by contradiction with $\psi([x]) = \mathcal{S}_{\sigma_{[x]}}$ for all $[x] \in \mathcal{X}/\sim_{\mathcal{G}}$.

Because $g_0 \neq e$, then $g_0 = g_\alpha$ for some angle $\alpha \in S \sim N(0, \sigma_{[x]})$ that is strictly positive or strictly negative. Assume $g_0 = g_\alpha$ with $\alpha > 0$. Because the sample $S$ is finite, there exists $u = max\{S\}$. Consider its corresponding element $s_u \in [x]_0$ w.r.t. the center of symmetry of the distribution as

$s_u = \rho_{\mathcal{X}}(g_u)c_{[x]_0}$. By equation 20,

$$\psi(s_u) = g_u \cdot g_\alpha = g_{u+\alpha}, \tag{21}$$

where $u + \alpha > u \implies \psi(s_u) \notin \mathcal{S}_{\sigma_{[x]_0}}$ as $u = max\{S\}$, which finalizes the proof. $\qquad\square$

**Proposition B.3.** *Consider a $\mathcal{G}$-invariant autoencoder $\delta \circ \eta$ and a group action estimator $\psi$. Under cyclic symmetries in $\mathcal{X}$, the following statements are equivalent:*

*(i) $\psi\left(c_{[x]}\right) = e \; \forall [x] \in \mathcal{X}/\sim_G$ for some $c_{[x]} \in [x]$.*

*(ii) $\forall [x] \in \mathcal{X}/\sim_{\mathcal{G}}$, the canonical representation of any $s \in [x]$ is the element $c_{[x]} \in [x]$.*

*(iii) $\forall [x] \in \mathcal{X}/\sim_{\mathcal{G}}$, it holds that $\psi\left([x]\right) = C_{n_{[x]}}$.*

*Proof.* The proof is immediate by following the proofs for *(i)* $\iff$ *(ii)* and *(i)* $\implies$ *(iii).* as in Prop. 3.1, substituting the uniformity of symmetries condition by cyclic condition. *(i)* $\implies$ *(iii)* is immediate. $\qquad\square$

### B.1 CONVERGENCE OF $L_2$ TOWARDS THE CENTERS OF SYMMETRY.

Minimizing $L_2$ intuitively encourages convergence towards the centers of symmetry of both uniform and Gaussian distributions, and in general, arbitrary symmetric distributions. Consider a class $[x] = \{\rho_{\mathcal{X}}(g)c_{[x]} \; s.t. \; g \in \mathcal{S}_{\theta_{[x]}}\}$ and the proposed minimization objective

$$\mathcal{L} = \mathcal{L}_1 + \mathcal{L}_2 = d_1\left(\rho_{\mathcal{X}}(\psi(x))\,\delta(\eta(x)), x\right) + d_2\left(\psi(x), e\right). \tag{22}$$

The standard IE-AE loss $L_1$ results in an arbitrary element of the orbit of $x$ chosen as canonical, that is, $\psi(o_x) = e$ for some $o_x \in \mathcal{O}_x$.

Now, focusing on minimizing the $\mathcal{L}_2 = d_2\left(\psi(x), e\right)$ loss in our model, it seems to merely ensure that the canonical representation for an IE-AE is an actual member of the equivalence class, meaning $\psi(s) = e$ for some $s \in [x]$. However, it can be shown that the specific member within this range that minimizes the $\mathcal{L}_2$ loss is in fact the center of this symmetry. To build on this intuition, consider the uniform case, and define the distance between two group elements $g_\alpha, g_\beta$ with rotation angles $\alpha, \beta \in [-180, 180]$ as $d_2(g_\alpha, g_\beta) = |\alpha - \beta|$. The objective $\mathcal{L}_2$ ensures that the canonical element is some element of the class i.e. $\psi(g_{\alpha_0}) = e$ for some $\alpha_0 \in [-\theta_{[x]}, \theta_{[x]}]$. Let us calculate the sum of the $L_2$ losses for all elements in the class:

$$\int_{-\theta}^{\theta} L_2(x)\,dx = \int_{-\theta}^{\theta} d(\psi(g_x), e)\,dx = \int_{-\theta}^{\theta} d(\psi(g_x), \psi(g_{\alpha_0}))\,dx =$$
$$= \int_{-\theta}^{\theta} |x - \alpha_0|\,dx = \int_{-\theta}^{\alpha_0} -x + \alpha_0\,dx + \int_{\alpha_0}^{\theta} -x + \alpha_0\,dx = \tag{23}$$
$$= \alpha_0^2 + \theta^2$$

This is minimized when $\alpha_0 = 0$, i.e. when the choice of the canonical is $\psi(g_0) = \psi(c_{[x]}) = e$.

The previous derivation can be similarly extended to the center of symmetry of Gaussian symmetries. In this context, we substitute the integral ($\int$) with a summation ($\sum$), considering that we are dealing with finite samples, to arrive to an equivalent understanding. Furthermore, this reasoning can be applied in general to arbitrary symmetric distribution, that is, any distribution that shows a center of symmetry.

As for the case of cyclic distributions, the result is straightforward, as the requirement for a center of symmetry is relaxed to $c_{[x]} \in [x]$, which is already sufficient to capture a cyclic distribution, given its inherently symmetric structure. The condition $c_{[x]} \in [x]$ is inherently satisfied by the $L_2$ minimization, which obviates the need for further derivations.

## C PSEUDO-LABEL ESTIMATORS

**Uniform Distributions** For simplicity in notation, we denote the set of rotation angles in $\psi(\mathcal{N}_k(x))$ by . Let's consider the distribution of these rotation angles, $\psi(\mathcal{N}_k(x))$, and its absolute value representation, $|\psi(\mathcal{N}_k(x))|$.

The decision to use the method of moments estimator for generating pseudo-labels is motivated by its robustness to outliers, compared to other potential estimators. Let's consider the alternative of

a maximum likelihood estimator (MLE) for the distribution $\psi(\mathcal{N}_k(x))$, which is the maximum in $\psi(\mathcal{N}_k(x))$. This approach is particularly sensitive to outliers. For example, if any element in $\mathcal{N}_k(x)$ has an erroneously predicted high rotation angle, the pseudo-label for $x$ will be disproportionately influenced by this outlier. In contrast, the method of moments estimator, which we employ, calculates the pseudo-label as two times the mean of all angles in $|\psi(\mathcal{N}_k(x))|$. This averaging effect mitigates the impact of any occasional anomalous angles, as these outliers are diluted when calculating the mean. Therefore, the method of moments approach offers a more stable and representative estimate.

Following this reasoning, we estimate the pseudolabels as $E{=}2 \cdot \overline{|\psi(\mathcal{N}_k(x))|}$, where $\overline{|\psi(\mathcal{N}_k(x))|}$ corresponds to the mean rotation angle in $|\psi(\mathcal{N}_k(x))|$. Additionally, when calculating $\mathcal{N}_k(x)$, we exclude the element $x$ from being its own nearest neighbour to avoid potential biases in the calculation of the pseudo-labels. Finally, we consider the optional implementation of an outlier detection method for $|\psi(\mathcal{N}_k(x))|$, such as the Interquartile Range (IQR) method. In our approach, rotations that are more than two standard deviations away from the mean are excluded, aiming to produce more consistent and stable pseudolabels.

**Gaussian Distributions** In the Gaussian case, $|\psi(\mathcal{N}_k(x))|$ follows a half-normal distribution. The standard deviation from the original normal distribution can be computed as $E = \frac{\sigma}{1-2/\pi}$ where $\sigma$ is the standard deviation of the sample $\psi(\mathcal{N}_k(x))$.

**Cyclic Distributions** We can estimate the cyclic group to which the distribution $\psi(\mathcal{N}_k(x))$ corresponds by calculating the Kullback-Leibler divergence from $\psi(\mathcal{N}_k(x))$ to $f_n$, where $f_n$ is the equivalent of $C_n$ as a continuous distribution defined in [-180,180].

To determine the cyclic group $C_n$ that best represents the distribution $\psi(\mathcal{N}_k(x))$, we compute the Kullback-Leibler (KL) divergence between $\psi(\mathcal{N}_k(x))$ and a continuous distribution $f_n$, defined as the continuous counterpart of the discrete cyclic group $C_n$ over the interval [180,180] degrees. Specifically, $f_n$ is constructed as a uniform distribution over $n$ equidistant points within this interval, reflecting the rotational symmetries in $C_n$:

$$f_n(\alpha) = \frac{1}{n} \sum_{i=1}^{n} \delta(\alpha - \alpha_i), \tag{24}$$

where $\delta$ is the Dirac delta function, $\alpha$ represents the angle in degrees, and $\alpha_i$ are the $n$ equidistant angles (symmetry positions) in [-180,180] corresponding to $C_n$. The KL divergence between the empirical distribution $\psi(\mathcal{N}_k(x))$ and $f_n$ is then calculated to quantify the similarity between them. This divergence is given by:

$$D_{\mathrm{KL}}(P\|Q_n) = \sum_i P(i) \log\left(\frac{P(i)}{Q_n(i)}\right) \tag{25}$$

where $P$ is the normalized histogram of $\psi(\mathcal{N}_k(x))$ over 360 bins and $Q_n$ is the discretized version of $f_n$ over the same bins as $P$.

This KL divergence metric allows us to estimate the most likely cyclic group $C_n$ that the distribution $\psi(\mathcal{N}_k(x))$ aligns with, providing a measure of how well the empirical distribution matches the expected rotational symmetries of $C_n$. Therefore, the pseudolabel of $x$ is calculated as:

$$\hat{\theta}_x = argmin_{n \in \mathbb{N}} D_{\mathrm{KL}}(P\|Q_n) \tag{26}$$

Practically, our comparisons are limited to a finite set of cyclic groups, ranging from $C_1$ to $C_8$, although the number of cyclic groups to consider can be increased. Finally, note that in order to be consistent with the notation established throughout the paper, we use the notation $\hat{\theta}_x$ for the pseudo-labels, although $\hat{\theta}_x$ belongs to $\mathbb{N}$ and represents the corresponding cyclic group, rather than a continuous value as in the uniform case.

# D ADDITIONAL EXPERIMENTAL INFORMATION

## D.1 MODEL CONFIGURATION AND TRAINING

In all our experiments, the encoder $\eta$, the network $\Theta$, and the constrained group function $\overline{\psi}$ are built using SO(2) equivariant/invariant networks from Weiler & Cesa (2019). We maintained consistent

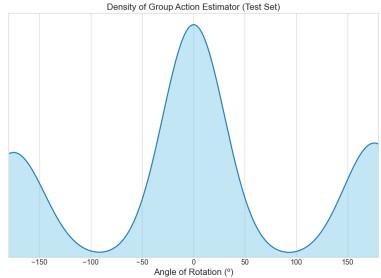

(a) Density of $\psi$ for digit 3 in MNISTC2C4.

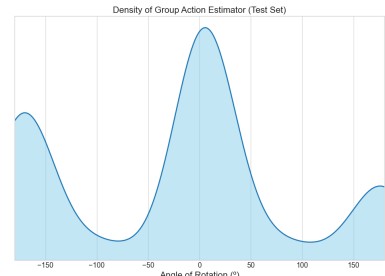

(b) Density of $\psi$ for digit 4 in MNISTC2C4.

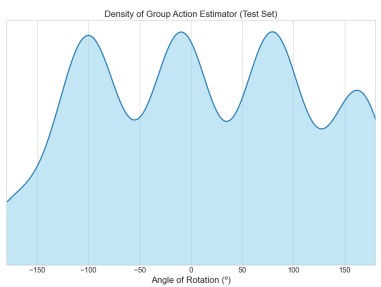

(c) Density of $\psi$ for digit 5 in MNISTC2C4.

Figure 7: Density of $\psi$ for MNSITC2C4 in different classes. It is important to note a visualization artifact near the x-axis limits at $-180°$ and $180°$. Due to these points lying exactly on the circular boundary (the break point of the circle $S^1$), the peaks appear artificially lower than their actual values in the plot. This apparent reduction in density is a result of projecting the circular distribution onto a straight line for visualization. In reality, both $C_2$ and $C_4$ distributions exhibit a single, prominent peak at the point where $-180°$ and $180°$ converge on the circle.

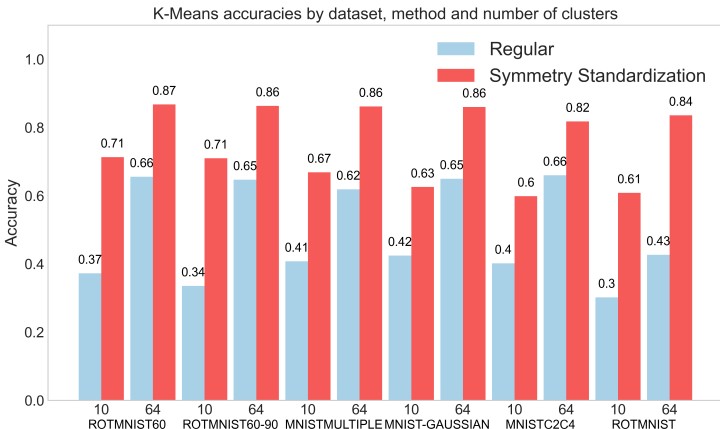

Figure 8: Results for K-Means.

network sizes across all experiments. The encoder architecture comprises seven SO(2) invariant convolutional layers with feature maps increasing from 128 in the first layer to 256, and 200 feature maps at the output. Each convolutional layer is followed by a batch normalization and a ReLU activation, except the final block which employs global average pooling. The $\overline{\psi}$ function resembles the encoder $\eta$ but employing SO(2)-equivariant convolutional layers with feature maps ranging from 64 to 128. Similarly, the $\Theta$ network resembles the group function $\overline{\psi}$ with 64 initial feature maps for continuous distributions, and 32 for cyclic distributions, followed by three fully-connected layers with ReLU activations. Lastly, the decoder $\delta$ is designed with conventional convolutional and upsampling layers to inversely replicate the $\eta$ encoder's structure. Note that the $\Theta$ network has a single neuron output for the uniform and Gaussian distributions case, while $n$ for the cyclic case, where $n$ is the number of cyclic distributions that we want to compare to. In our case, we use $n = 8$, but arbitrarily higher order groups can be considered.

The number of neighbors $k$ for the $\Theta$ network varies with each experiment: 45 for uniform and Gaussian distributions, and 150 for cyclic. Cyclic group estimations require more neighbors as this computation relies on the comparison between distributions via the KL divergence, which demands a higher number of points to be reliable. In contrast, continuous distributions allow fewer neighbors, as their pseudo-labels are derived through parameter estimation, typically reliable with $k$ over 30. Selecting the number of neighbors involves balancing better accuracy against the risk of incorporating elements with differing symmetry distributions, which could destabilize the pseudo-label estimations.

During the pre-training phase and self-supervised training, the models undergo 400 and 150 epochs respectively. Both the constrained IE-AE and the $\Theta$ network use the Adam (Kingma & Ba, 2017) optimizer, combined with a cosine scheduler with 5 warm-up epochs. The learning rates are set at 0.01 for the IE-AE and 0.001 for the $\Theta$ network. Additionally, the $\mathcal{L}_2$ loss is assigned a weight of 0.03125 to maintain balanced optimization in conjunction with the $\mathcal{L}_1$ loss.

### D.2 OUT-OF-DISTRIBUTION AND SYMMETRY STANDARDIZATION EXPERIMENTS

For the out-of-distribution symmetries detector, we use the models obtained after training in the MNIST variations with partial symmetries. No training is necessary, as the classifier relies on the generalization capabilities of the $\Theta$ function, already trained. During inference, an input is classified to be out of the distribution of the training dataset when its predicted group action angle $\psi(x)$ is outside of the symmetry distribution predicted by $\Theta$. In the case of cyclic distributions, because the predicted distributions in $\Theta$ are discrete, we consider that an input is out-of-distribution if it deviates more than $5°$ from an element of $C_n$. For the symmetry standardization, baseline supervised and unsupervised models are first trained and tested in the datasets variants to create the "no symmetry standardization" results. Similarly, the "symmetry standardization" are created using the symmetry standardized training and test datasets obtained after training our model. K-Means is trained with different number of clusters as shown in Fig. 8. Classification of each of K-Means is calculated based on class majority of that cluster. ResNet-18 is trained from scratch for 100 epochs, using an

Adam optimizer with learning rate 0.001. Finally, a KNN supervised classifier with 5 neighbors is trained on the $G-$invariant embeddings of the standard, pre-trained IE-AEs in each of the datasets. Test accuracy is computed similarly, by first computing the IE-AE embeddings and then predicting with the trained KNN.

Table 3: Mean predicted level of symmetry for symmetry prediction in test set for each dataset.

| Class | MNISTRot60 | | MNISTRot60-90 | | MNISTMultiple | | MNIST | | MNISTRot | | MNISTGaussian | |
| | $\theta$ | $\overline{\Theta}$ | $\theta$ | $\overline{\Theta}$ | $\theta$ | $\overline{\Theta}$ | $\theta$ | $\overline{\Theta}$ | $\theta$ | $\overline{\Theta}$ | $\sigma$ | $\overline{\Theta}$ |
|---|---|---|---|---|---|---|---|---|---|---|---|---|
| 0 | 60º | 61.04º | 60º | 65.18º | 0º | 8.66º | 0º | 8.22º | 180º | 177.29º | 0º | 15.23º |
| 1 | 60º | 63.14º | 60º | 63.18º | 18º | 4.87º | 0º | 1.79º | 180º | 179.27º | 9º | 8.06º |
| 2 | 60º | 61.33º | 60º | 67.66º | 36º | 35.26º | 0º | 6.25º | 180º | 178.23º | 18º | 23.69º |
| 3 | 60º | 60.45º | 60º | 66.30º | 54º | 58.85º | 0º | 4.14º | 180º | 179.93º | 27º | 27.25º |
| 4 | 60º | 59.44º | 60º | 64.90º | 72º | 75.70º | 0º | 6.57º | 180º | 181.55º | 36º | 38.89º |
| 5 | 60º | 63.33º | 90º | 85.46º | 90º | 88.48º | 0º | 8.22º | 180º | 184.55º | 45º | 42.46º |
| 6 | 60º | 60.64º | 90º | 85.05º | 108º | 106.83º | 0º | 6.68º | 180º | 178.53º | 54º | 50.35º |
| 7 | 60º | 60.60º | 90º | 84.88º | 126º | 117.91º | 0º | 3.12º | 180º | 181.33º | 63º | 53.02º |
| 8 | 60º | 59.28º | 90º | 82.78º | 144º | 134.08º | 0º | 4.14º | 180º | 175.30º | 72º | 59.34º |
| 9 | 60º | 61.55º | 90º | 83.66º | 162º | 156.38º | 0º | 4.48º | 180º | 179.65º | 81º | 64.70º |

Table 4: Mean Absolute Error of symmetry level prediction in the test set for each dataset.

| Class | MNISTRot60 | | MNISTRot60-90 | | MNISTMultiple | | MNIST | | MNISTRot | |
| | $\theta$ | MAE | $\theta$ | MAE | $\theta$ | MAE | $\theta$ | MAE | $\theta$ | MAE |
|---|---|---|---|---|---|---|---|---|---|---|
| 0 | 60º | 4.92 | 60º | 5.09 | 0º | 25.62 | 0º | 5.43 | 180º | 8.38 |
| 1 | 60º | 9.51 | 60º | 7.96 | 18º | 8.19 | 0º | 0.53 | 180º | 6.64 |
| 2 | 60º | 4.00 | 60º | 6.24 | 36º | 11.63 | 0º | 2.75 | 180º | 7.94 |
| 3 | 60º | 3.72 | 60º | 5.78 | 54º | 8.88 | 0º | 1.56 | 180º | 7.98 |
| 4 | 60º | 4.68 | 60º | 6.76 | 72º | 12.51 | 0º | 2.20 | 180º | 9.00 |
| 5 | 60º | 5.94 | 90º | 5.43 | 90º | 10.43 | 0º | 1.98 | 180º | 7.98 |
| 6 | 60º | 3.86 | 90º | 5.09 | 108º | 11.17 | 0º | 3.16 | 180º | 9.27 |
| 7 | 60º | 3.62 | 90º | 7.87 | 126º | 14.95 | 0º | 1.33 | 180º | 8.90 |
| 8 | 60º | 4.19 | 90º | 7.27 | 144º | 14.48 | 0º | 1.46 | 180º | 11.78 |
| 9 | 60º | 4.67 | 90º | 5.81 | 162º | 18.46 | 0º | 1.41 | 180º | 10.05 |

