# OpenReview forum: "Self-Supervised Detection of Perfect and Partial Input-Dependent Symmetries"
_ICLR.cc/2024/Conference — Submitted to ICLR 2024_

### Official Review · Reviewer_JVYE · 2023-10-30

**Soundness:** 3 good
**Presentation:** 3 good
**Contribution:** 2 fair
**Rating:** 6
**Confidence:** 4

**Summary:**

This work build upon the IE-AE architecture, enhancing it so that the distribution of observable group elements (assumed uniform) is learnt. This is effective when only partial symmetries exist in the data, even in the case where each class has a different group element distribution acting on it. This work focuses solely on the SO(2) group (continuous rotations). An additional network $\Theta$ that predicts the rotation is included on top of IE-AE, as well as a strategy based on Nearest-Neighbors to estimate the boundary of the uniform distribution.

The overall method is very well presented, and nicely backed with theory. The experiments on MNIST-like datasets also back the theory up, showing accurate learning of the symmetry boundaries.

**Strengths:**

* The theoretical derivation is sound and solid.

* The improvement over IE-AE seems reasonable from an architectural point of view.

* The MNIST-like data experiments nicely show the properties of the method.

**Weaknesses:**

* Although the spirit of this work is fundamentally setting the grounds of the technique and the theory behind it, I believe experiments should explore data beyond MNIST. In that sense, I would strongly suggest the authors to explore (at least) CIFAR-like data, with 32x32 RGB images.

* There are some unanswered questions related to data that is inherently invariant to SO(2). For example, the Flowers dataset contains top-down photos of flowers, known to be almost invariant to rotation. Some experiment showing the limitations of the method (in this sense or another) would be extremely welcome.
  * In a similar vein, another way to show the limitations of the method (and strengths hopefully!) is to perform an experiment on data that does not strictly obey a uniform distribution of group elements.

* The work lacks an ablation study, given that several losses are combined, and there exist additional parameters such as $k$. Additionally, in the appendix there is mention to an additional consistency loss that is not mentioned in the main body, which I think should be amended for scientific rigour.

**Questions:**

* The estimation of the symmetry boundary using $E=2 \overline{|\psi(N_k(x))|}$ is tailored for distributions of the form $U(-\theta,\theta)$. Have the authors thought about generalizing the method to arbitrary distributions? Even if it is only the case of arbitrarily bounded uniform distributions. In such case, the concept of "center of symmetry" as explained in the paper would not be valid anymore. Although uniform symmetry distributions are sensible, multimodal ones could naturally arise in datasets too.
     * Related to this question, one experiment that would be interesting is to have non-overlaping rotation intervals per class (instead of all being subsets of the next class as in MNISTMultiple).

* As far as I understand, the center of symmetry is an abstraction for "canonical rotation", which could be any absolute angle in practice. Therefore, the proposed method cannot estimate the absolute rotation of an image, but only the rotation wrt. its canonical one. Is this correct?
  * What is the role of $L_2$. In general, it pushes the estimation given by $\psi$ to be concentrated around $e$, right? Is the underlying idea to use it as some sort of regularization?

* How limiting is the assumption _"we shift from using equivalence classes $[x]$ to sets of objects semantically similar to $x$ for estimating $\theta_x$"_. Would it be possible to find the equivalence classes in practice? Or this is a form that helps the theoretical aspect of the method only?

* Substituting the equivalence class by the $k$-Nearest Neighbors might rely strongly on the NT-Xent loss being satisfied. How important is the NT-Xent loss? I think an ablation of the different losses that compose the final loss would be of interest for this work, so readers understand the contributions and how to practically use the proposed approach.
  * Additionally, NT-Xent usually requires quite large batch sizes. What is the training setting in this case? How are the positive and negative pairs formed?
  * Is there a way to bound the boundary estimation error as a function of $k$?

* In the Appendix I read: _"An additional consistency loss is trained to ensure that the representations created by the contrastive routine and the reconstruction routine are similar and do not conflict each other."_
  * Can the authors elaborate on this? If there is an additional loss that helps improve the results, I would suggest to include it in the main body of the paper. This adds another weighing parameter to the overall loss, which reinforces the need of an ablation study in that sense.

* What do the up-down arrows from $\theta$ to $\hat{\theta}$ represent in Figure 3? I suggest replacing them with "MSE" or $L_3$. Additionally, writing "supervised learning" might mislead some reader. It is supervised, but with pseudo-labels, which is a much less strong condition of the method (no need for annotations).
  * Additionally, I suggest the authors to highlight in Figure 3 where the proposed losses are applied. For example $L_1$ takes the left and right $X$s. Loss $L_3$ takes $\theta$ and $\hat{\theta}$, etc.

* About results in Table 2 (Improving non-equivariant models with symmetry standardization). I notice that the improvement becomes smaller as the range of rotation is smaller. What is the intuition behind that effect? In that Table, how does the proposed method compare with a method that infuses equivariance without knowledge of distribution like IE-AE? I think this comparison is required, to understand whether the improvements shown are significative wrt. SOTA.

* How does the method behave for images inherently invariant to rotation (eg. a circle). This would be the case of the Flowers dataset for example, where many flowers are photographed top-down and appear centered and "almost" invariant to rotation.

* I suggest adding [Suau et al. ICML 2023](https://arxiv.org/abs/2306.16058) given some similarities in the spirit of the work. In such paper, equivariances are learnt from data, and the distribution over $g$ elements is naturally learnt.

----
**Overall comment:**

The paper is very well written, almost typo-less, and with excellent language. I enjoyed the reading, I thank the authors for that. The theoretical formulation is solid, and the proposed method to learn symmetry boundaries is novel to the best of my knowledge. However, I still think this paper could strongly benefit from experiments with data beyond MNIST, as well as a thorough ablation study, as mentioned before. I also suggest to add some explicit discussion about the strengths and limitations of the method.

---

> ### Author Response · Authors · 2023-11-23
> **Reviewer JVYE**
>
> We thank the reviewer for their appreciation of our paper, careful reading and constructive feedback towards improving our manuscript. We address the reviewer’s points below (numbered in order) and have updated our manuscript accordingly:
>
> (1) **Although the spirit of this work... In a similar vein... The work lacks an ablation study...**
>
> +->  We updated our manuscript showing how our proposed framework can be applied to different distributions such as Gaussian or cyclic, and more generally, arbitrary (unimodal) symmetric distributions. We have been on a time crunch to extend our method to other distributions and groups, but we believe that our method can be extendable to other groups such as O(2) and to 3D rotations, and we are working on having these experiments on more realistic datasets for the camera-ready version. We also thank the reviewer for suggesting to test our model on data that does not strictly follow a uniform distribution, which resulted in new results as reflected on the updated manuscript. Lastly, the use of the NT-Xent loss has been removed from our method, as the improvements have proved to be marginal or non-existent after further hyperparameter optimization of the rest of the parts of the method. Therefore, no ablation study on this loss and the consistency loss is needed anymore, as both are no longer part of the method. Regarding CIFAR10, While the method promises to be applicable to other datasets such as the ones tested in the IE-AE work, in datasets with high intra-class variability e.g. CIFAR10, objects belonging to the same class may not share a G-invariant representation that connects them through a group action. In such cases, the standard IE-AE is unable to capture this relationship, and consequently, our method is also limited, as it relies on IE-AEs. We have included this discussion in the limitations section of our method (Section 6).
>
> (2) **There are some unanswered questions…** In the context of IE-AEs, the treatment of inherently symmetrical data can be predicted through the theory underlying these models. An IE-AE determines group transformations by using the function $\psi$ to rotate the G-invariant canonical representation back to the original input's orientation. For an input that possesses inherent full rotational invariance, any rotation applied to its canonical representation will align with the original input's orientation, therefore any prediction by $\psi$ is valid as a solution. This is because the input itself is invariant under all rotations in the group G. Thus, the predicted transformation by $\psi$ in such cases can be the identity (0º) or any rotation at all, as the input's inherent symmetry makes every rotated version identical to the original. This prevents an IE-AE based method to discern whether a shape is inherently invariant to all rotations.
>
> Note that when asking 'What is the level of rotational symmetry of this inherently invariant input in the dataset?' there are multiple valid answers as per the previous reasoning: the symmetry level could be described as zero, or encompassing the full SO(2) group, or actually any other distribution within SO(2), as any of the choices maintain the given input invariant in the dataset. In the case of our IE-AE, the $L_2$ term will push $\psi$ towards a degenerate symmetry distribution in 0º for this kind of inputs.
>
> (3) -> Understanding the functioning of $L_2$ and how it leads to centers of symmetry in the canonical representation is important to address the raised concerns that follow. At first sight, minimizing $L_2$ in combination with the IE-AE loss seems to only ensure that the canonical representation of an equivalence class is a member of that class, as opposed to the standard IE-AE approach where the canonical representation is chosen arbitrarily. However, it can be proved that the element of the class that minimizes $L_2$ is in fact the center of symmetry of the class. This derivation is included in the updated manuscript (see Appendix B.1). Now let's connect this reasoning to your questions.

---

> ### Author Response · Authors · 2023-11-23
> **Continuation**
>
> (3.1) **The estimation of the symmetry…**
>
> +->  For multimodal distributions there is indeed no center of symmetry, but several, and the reasoning in Appendix B.1 seems to suggest that minimizing the $L_2$ objective would likely lead to capturing the median point of the multi-modal distribution as a canonical representation. This could, in principle, be enough to generate pseudolabels to estimate several sub-centers of symmetry with a well designed pseudolabel estimation method. Although we did not get to research the multi-modal case, we believe that with the recently added extension of our method to unimodal, symmetric distributions and cyclic groups, our work sets a solid and flexible theoretical framework for detecting many types of symmetry distributions, which in principle could include multi-modal ones.
>
> Regarding the experiment about non-overlapping rotation intervals per-class, note that the unsupervised nature of the problem makes the experiment that you propose precisely the same as the ones presented in our paper. To understand this, let’s say that instead of rotating the MNIST class 3 in [-20º, 20º] and the class 4 in [-40º,40º], we rotated the class 3 in [80º, 120º] instead. Because the data is unsupervised, the network is not aware of what a ‘straight’ digit 3 is. However, our method realizes that digits of such a form have rotational symmetries around some center of symmetry, which in this case is the ‘straight’ digit 3 rotated in 100º. Therefore, even though we have rotated the class 3 in [80º, 120º] instead of in [-20º, 20º], our method will still detect that the level of symmetry of the input is described by a  [-20º, 20º] whose center of symmetry is on the 100º orientation. In other words, different non-overlapping per-class intervals will be transformed into overlapping ones due to the ability of our method to find centers of symmetry i.e. to establish a natural reference point in 0º. This means that, for instance, the MNISTMultiple experiment is equivalent (or even exactly the same) as the one you propose. We hope that this explanation answers the question.
>
> (3.2) **As far as I understand...**
>
> +-> Indeed, the rotations predicted are with respect to a canonical representation, but we believe it is necessary to establish such a reference point when detecting levels of symmetry. This task inherently involves quantifying how much an input deviates from a state of non-symmetry, which we define as the identity element of the group (e.g. a 0º rotation in SO(2)) or, more generally, a meaningful canonical representation. Establishing such a reference point is crucial, even in scenarios with multimodal distributions, in which the natural choice for canonical representations would be the center of symmetry of each sub-mode within the multimodal distribution.
>
> (3.3) **How limiting is...**
>
> +-> In practice, equivalence classes could be approximated by using large enough discretizations of SO(2), but this approach would be computationally unfeasible. This approach also has the disadvantage that, under some distance measures, the computed class would be degenerate in many cases. For instance, if in this hypothetical computation of equivalence classes, we decide to compare elements of the dataset pixel-by-pixel in order to discern if they are in the same class, we would need rotated versions of each input to be exactly the same at each pixel value. We could of course define an acceptable threshold beyond which we include an element in an equivalence class, but this still would not solve the need for lots of compute for a large-enough discretization of SO(2) in large datasets and/or images. In contrast, if instead of comparing the images  pixel-by-pixel, we consider comparing the cosine similarity of the embeddings in the latent space, it is more likely that our set will capture the symmetries of the data without running into degenerate sets. In summary, a purely equivalence-class based method could be approached, but we believe that using the G-invariant latent space to estimate the equivalence classes is much more elegant and robust. Rather than limiting, our assumption makes our model more flexible in realistic scenarios, as bypassing the equivalence class calculation makes the method considerably cheaper, and its consideration is grounded in a very realistic assumption. The main limitation about this shift is that if the assumption turns out not to be true in a particular case, then the set $N(x)$ set will not be representative of the symmetries of $x$. However, it is incredibly difficult to come up with scenarios, both realistic and synthetic, in which similar inputs don't share symmetries.

---

> ### Author Response · Authors · 2023-11-23
> **Continuation**
>
> (4) **Substituting the equivalence class... In the Appendix I read...**
>
> +->  The symmetry level error depends on various factors such as the quality of the embeddings in $L_{inv}$, or the extent of class imbalance within the data. For example, if a class is significantly underrepresented in the dataset – say, with only 20 samples – the predictions for the symmetry level of this class are likely to be less reliable. We do not think that it is possible to give a bound for this error, as it depends on several difficult-to-quantify factors, such as class imbalance in unsupervised settings or the quality of the embeddings.
>
> (5) **What do the up-down arrows...**
>
> +-> The up-and-down arrows aim to represent supervised learning between the prediction and the pseudo-label. We included the suggested changes in Figure 3 for clearer understanding. Please refer to the new manuscript.
>
> (6) **About results in Table 2...**
>
> +->  The effectiveness of symmetry standardization becomes more pronounced with the increased presence of rotational symmetries in the dataset. Non-equivariant models face significant challenges when dealing with data exhibiting extensive symmetries. Symmetry standardization, by collapsing elements into their centers of symmetry, effectively simplifies the modeling task for these non-equivariant models. Hence, the benefits of symmetry standardization are most apparent in scenarios with abundant symmetries. We included in the revised manuscript an extended discussion and comparison with the invariance achieved through IE-AEs.
>
> ---
>
> We hope that these responses clarify your questions and concerns. Please let us know if you have any follow-up / additional questions.
>
> Best regards,
>
> The Authors

---

### Official Review · Reviewer_d1fY · 2023-11-01

**Soundness:** 2 fair
**Presentation:** 3 good
**Contribution:** 2 fair
**Rating:** 5
**Confidence:** 3

**Summary:**

This paper introduces a method that predicts the distribution of symmetries of each input in a dataset. Building on the invariant-equivariant autoencoder (Winter et al., 2022) and partial group equivariant CNNs (Romero & Lohit, 2022), the proposed method is designed to detect the symmetry boundary of image rotations. Experiments on various rotated MNIST datasets show that the proposed method is effective in predicting symmetry levels, detecting samples with out-of-distribution symmetries, and improving non-equivariant models with symmetry standardization.

**Strengths:**

-	This paper implements the novel idea of capturing levels of symmetry at a sample-level, in contrast to previous partial equivariant models that captures levels of symmetry at a dataset level.
-	The proposed method can learn meaningful and consistent canonical representations defined by the center of symmetry. This is an improvement over (Winter et al., 2022), where the canonical representation depends on various factors in training.
-	The experiment demonstrates the effectiveness of the approach of detecting input-dependent symmetries. Specifically, the proposed method is able to predict symmetry levels, detect out-of-distribution symmetries, and standardize datasets to improve non-equivariant models.

**Weaknesses:**

-	While the proposed method seems general enough to process any symmetry, the only setting discussed is learning symmetry boundary of SO(2). Moreover, the assumption that the rotation symmetry is uniformly distributed in an interval seems restrictive.
-	It is not clear whether or to what extend different levels of symmetry exists in datasets. In the abstract, the authors motivated the task of detecting the level of symmetry by the example that pictures of cars and planes exhibit different levels of rotation in CIFAR-10 dataset. However, this difference is not quantified.
-	While experiments in the paper demonstrate that the proposed method works as intended, it is not clear how these tasks are relevant in real-world applications. The practical value of detecting symmetry levels could be made more convincing by discussing possible scenarios where predicting levels of symmetry gives useful information or is expected to improve downstream model performance.
-	Since the proposed method builds on invariant-equivariant autoencoder, comparing against it in experiments would help demonstrate the advantage of predicting input-dependent levels of symmetry, particularly for the out-of-distribution symmetry detection and improving models with standardization experiments. Currently there is no baseline comparison for any of the experiments.
-	Minor issue: the legends in figure 6 could be larger.

**Questions:**

-	Suppose a symmetry is outside the symmetry boundary, i.e. does not present in the training set. Do we want the neural network to be equivariant to it? If so, it seems detecting symmetry at a sample-level is less useful in avoiding over constraining the model. If not, doesn’t this limit the model’s generalization ability?
-	In section 3.2, why does minimizing $d(\psi(x),e)$ encourage $\psi(c_{[x]})=e$?

---

> ### Author Response · Authors · 2023-11-23
> **Reviewer d1fY**
>
> We thank the reviewer for their careful reading and constructive feedback towards improving our manuscript. We address the reviewer’s points below (numbered in order) and have updated our manuscript accordingly:
>
> 1 **While the proposed method seems general enough...**
>
> +-> We updated our manuscript showing how our proposed framework can be applied to different distributions such as Gaussian or cyclic, and more generally, arbitrary (unimodal) symmetric distributions. We hope that this update shows that our method is general enough to be compared with other methods such as the standard IE-AE. We recognise that we have been on a time crunch to extend our method to other distributions and groups, but we believe that our method can be further extendable to other groups such as O(2) and to 3D rotations, and we are working on having these experiments for the camera-ready version.
>
> 2 **It is not clear whether or to what extend different levels...**
>
> +-> The comparison between cars and planes intended to serve as an intuitive example of two objects that can appear with different symmetry levels in the same dataset, but we indeed did not quantify this difference in CIFAR10. The purpose of this is to offer an intuitive example of two objects that would usually appear with different symmetries: cars, showing very little rotational symmetries and planes, which when photographed from below, show full rotational symmetries.
>
> 3 **While experiments...**
>
> +-> In Section 5, we presented some of the practical applications of detecting levels of symmetry: building out-of-distribution symmetry classifiers and using symmetry standardization as a pre-processing tool to improve the generalization and performance of non-equivariant models. The benefits of learning the levels of symmetry were further motivated by the works presented in the Introduction section.
>
> 4 **Since the proposed method builds on invariant-equivariant autoencoder...**
>
> +-> Comparison of the IE-AE with our method is included now in the updated manuscript for the symmetry standardization experiments (Section 5). Note that a comparison with the IE-AE for the out-of-distribution symmetry experiment can not be performed, as this type of practical application can only be obtained by jointly using the $\Theta$ network and the modified IE-AE. In effect, a regular IE-AE can not be used to build out-of-distribution symmetry detectors, as it lacks the ability to predict an input-dependent symmetry distribution to compare each input to. Lastly, the figures of the symmetry level predictions in Section 5 have been modified for improved readability, as suggested.
>
> 5 **Suppose a symmetry is outside the symmetry boundary...**
>
> +-> As mentioned in our introduction, prior research suggests that full equivariance can overly constrain models, leading to reduced performance when the symmetry group defined for G-equivariance does not align with the actual symmetries present in the data. However, models exhibiting full equivariance demonstrate enhanced robustness when encountering test data with symmetry distributions that differ from the training data (e.g. an SO(2)-equivariant network trained on MNIST will have higher accuracy in RotMNIST during inference than a regular CNN trained on the same dataset). Conversely, these fully equivariant models may underperform compared to partially equivariant models, when exposed to test data that falls within the symmetry constraints of the training data (which is usually the case). In such cases, partially equivariant models are desired. This presents a trade-off: generalization to out-of-distribution symmetries versus optimized performance for in-distribution symmetries. Interestingly, combining our out-of-distribution symmetry detectors with both fully and partially equivariant models could be a promising approach. This might enable a model to adaptively balance between full and partial equivariance, leveraging the benefits of both for improved generalization and performance. We hope that this paragraph answers your question, and we thank you for raising an interesting point that poses a valuable addition to our manuscript and future work.

---

> ### Author Response · Authors · 2023-11-23
> **Reviewer d1fY**
>
> 6 **In section 3.2, why does minimizing...**
>
> +-> This is an excellent question. It can be somewhat tricky to understand why minimizing our proposed objective function $L_2 = d(\psi(x), e)$ results in fulfilling the constraint  $\psi(c_{[x]})=e$. At first sight, minimizing $L_2$ in combination with the IE-AE loss seems to only ensure that the canonical representation of an equivalence class is a member of that class, as opposed to the standard IE-AE approach where the canonical representation is chosen arbitrarily. However, it can be proved that the element of the class that minimizes $L_2$ is indeed the center of symmetry. This derivation is included in the updated manuscript (see Appendix B.1). As the new derivation shows, the objective $L_2$ is able to create the right conditions for symmetry level prediction, for any unimodal, symmetric distribution. Our method is not limited to this class of distributions only, as the same minimization objective can also lead to predict cyclic symmetry levels (as presented in the revised manuscript) and possibly other groups and distributions. We are working on having these experiments for the camera-ready version.
>
> ---
>
> We hope that these responses clarify your questions and concerns. Please let us know if you have any follow-up / additional questions.
>
> Best regards,
>
> The Authors

---

### Official Review · Reviewer_9nMD · 2023-11-01

**Soundness:** 2 fair
**Presentation:** 1 poor
**Contribution:** 1 poor
**Rating:** 3
**Confidence:** 2

**Summary:**

The paper proposes a method to solve for the classification task where the dataset consists of varying degrees of rotation for each class. The method consists of two steps: a) breaking down the input into an invariant and an equivariant component. b) Applying Partial G-CNN in the IE-AE architecture to capture the partial symmetries c) Applying self-supervised objective over it.

**Strengths:**

The problem statement of capturing perfect+partial symmetries is well motivated.

**Weaknesses:**

- Method section writing could be simplified, by taking rotation detection as a running example, and relating it with the notations intermittently.
- Clustering results on the output of the pseudo-labels and the final result from the contrastive learning approach would be beneficial in understanding the method.
- The novelty in the method is lacking as it builds up on the IE-AE with existing Partial G-CNN layers, coupled with contrastive learning approach. All of which are already existing in the literature.
- Motivation for applying contrastive learning is unclear, why can’t pseudo labels be directly used to predict the result? What is the benefit of applying contrastive learning?

**Questions:**

- First and third rows in Fig. 4 are same. What is the reason behind it? Is it the input?

---

> ### Author Response · Authors · 2023-11-23
> **Reviewer 9nMD**
>
> We thank the reviewer for their careful reading and raised concerns. We address the reviewer’s points below (numbered in order) and have updated our manuscript accordingly:
>
> 1 **Method section writing could be simplified...**
>
> +-> We recognise that it can be challenging to follow the definitions without a running example. Some of the most crucial terms defined in the method section now show a running example, in order to make the presented results easier to understand (please refer to the updated method section in the manuscript, new changes in the manuscript are coloured in red).
>
> 2 **The novelty in the method is lacking...**
>
> +-> The method builds on top of the IE-AE and modifies it to provide it with new, desirable properties that are necessary to learn the level of symmetry of each input, as stated in Proposition 3.1. Partial G-CNNs layers are not used in any part of the method, but rather motivate both the need for determining the level of symmetry, as well as the approach of modelling the level of rotational symmetry as a single-parameter estimation of uniform rotations centered at the identity. This nuance is now made clearer in the updated manuscript (Section 2.3). Similarly, the use of contrastive learning is optional, does not contribute towards the novelty of the proposed method, and has been removed in the revised manuscript, as we found the improvements to be marginal after further hyperparameter optimization of all the proposed components.
>
> 3 **Motivation for applying contrastive learning is unclear... What is the benefit of applying contrastive learning?**
>
> +-> The motivation behind contrastive learning is that minimizing a contrastive objective gives greater separation in the latent space, which is useful to create higher quality neighbors and improve the accuracy of the proposed method. However, the application of contrastive learning is not necessary for the method to be able to calculate the level of symmetries. Importantly, the use of the NT-Xent loss has been removed from our method, as pointed out in the previous response.

---

> ### Author Response · Authors · 2023-11-23
> **Continuation**
>
> 4 **First and third rows in Fig.4 are same...**
>
> +-> First and third rows are indeed inputs, while the rows underneath represent each input’s reorientation into their canonical representation. This figure is now modified to clearly define and name each row.
>
> ---
>
> We appreciate the reviewer’s feedback and have made comprehensive revisions to our manuscript. We believe these updates significantly enhance the clarity and understanding of our method, and hope that the reviewer finds the revised sections reflective of these improvements and more aligned with the objectives of our work. Please let us know if you have any follow-up / additional questions.
>
> Best regards,
>
> The Authors

---

### Official Review · Reviewer_9t9K · 2023-11-03

**Soundness:** 3 good
**Presentation:** 3 good
**Contribution:** 2 fair
**Rating:** 5
**Confidence:** 4

**Summary:**

This work suggests a novel method for detecting symmetries in a given dataset.
It is done in several stages, in an usupervised manner, and is able to estimate the distribution of symmetries that is shared by each subset of items that form an equivalence-class, under the specific group action. This allows a cannonical normalization of the data, which can be desirable for further processing.
The method is based on a proposition that shows the equivalence of different properties of the group action estimator for an equivalence-class of items. Theory, analysis and experiments are under the setting of image data with uniform distributed rotation symmetries over a continuous segment of angles. The method is demonstrated on several variants of the MNIST dataset, which contained controlled syntetic ranges of rotations. Experiments focus on prediction of the per-item distribution parameter (size of range), but also show improvements obtained by both supervised and unsupervised classifiers after symmetry-standardizing the data, using the proposed method.

**Strengths:**

1] The choice to tackle a rather challenging combination of several main difficulties (that were previously only tackled in separation): (i) Being unsupervised ; (ii) Dealing with partial (versus full group action) symmetries (iii) Allowing per item level of symmetry. Each of these components of the setting are important requirements towards the ability to tackle real data in a general way, and it is a very important goal, in my opinion, to make progress in this difficult setting.

2] The method is backed up by a very formal framework, of partial symmetry groups acting on items of the dataset, in a way that connects to previous work and gives a general perspective on the problem, under which very specific choices and assumptions have been made. This formulation gives hope that the method could be extended to other, more general, related settings.

3] The experiments show that the method can pretty well predict the desired symmetry parameters, at least for the rotation variants of the MNIST dataset.

**Weaknesses:**

1] While the decision to generalize the setting of previous works (especiall w.r.t. Partial-G-CNNs and IE-AE) to deal with partial, per-item distributions, in an unsupervised manner, the result is much more limited and less applicable as a result:
* The assumption of pure rotation, that is uniformly distributed over a single segment (practically a 1-parameter estimation problem) is very limited and unnatural. This can be compared to IE-AE that handle discrete and continuous groups, including rotations, translations and permutations.
* This limited setting is very obvious in the testing, which includes only uniformly rotated versions of MNIST. The above mentioned methods include demonstrating the potential on some slightly more realistic datasets (such as CIFAR and Shapenet), where there is no control or no way to model the true underlying symmetries. Especially when considering the potential of unsupervised learning, it is a little disappointing to see it demonstrated on controlled data that could allow fully supervised training.
* There is not much of a discussion as to how you could drop some of the assumptions, if desired. For example, the uniformity is currently what enables rather easily estimating the center of the range and its boundaries. What would happen with other types and distributions of symmetries?

2] The need for  (and the quality of) the self-supervision and the boundary prediction network $\Theta$ itself is unclear. What I mean here is that it is not the case that the network $\Theta$ can generalize to a new dataset, so for the given dataset, you still have to run the pre-training and then, given a new sample, you could just predict the pseudo-label itself. I imagine that it would be more precise, but perhaps less efficient because of the nearest neighbor computation. This should further explained.

3] The way of calculating nearest neighbors is not specified (or perhaps I missed it). This is critical for identifying the equivalence classes.

**Questions:**

1] How are nearest-neighbors ("semantically similar inputs") found?
2] Generating the pseudo-labels: Why is the estimator that simply takes the mean angle considered to be robust? and in particular "more robust to outliers" than what?
3] Experiments: Have you tried to demonstrate the method on more realistic data? (which would perhaps require dropping the uniform or continuous assumptions, or perhaps trying other more general classes of symmetry).

---

> ### Author Response · Authors · 2023-11-23
> **Reviewer 9t9K**
>
> We thank the reviewer for their careful reading and constructive feedback towards improving our manuscript. We address the reviewer’s points below (numbered in order) and have updated our manuscript accordingly:
>
> 1 **The assumption of pure rotation..**
>
> +->We updated our manuscript showing how our proposed framework can be applied to different distributions such as Gaussian or cyclic, and more generally, arbitrary (unimodal) symmetric distributions. We hope that this update shows that our method is general enough to be compared with other methods such as the standard IE-AE.
>
> 2 **This limited setting is very...**
>
> +-> In our work, we followed experimental settings set by previous works such as the IE-AE, which used controlled symmetries in datasets like RotMNIST, Tetris Shape dataset with random SO(3) rotations, and Shapenet with random 90º rotations. Indeed, we recognize the importance of applying our method to more complex datasets such as the ones in this work, and the truth is that time constraints have limited the extent of our method in this initial phase, both regarding its extension to other distributions and experiments on more datasets. We believe that our method can be extendable to distributions in other groups such as O(2) and 3D rotations, and we are working on having these experiments for the camera-ready version.
>
> 3 **There is not much of a discussion as to how you could drop...**
>
> +-> As mentioned in the first response, it is indeed not necessary to have a uniform distribution in the data. The main assumption of the method is the existence of centers of symmetry, which enables our model to detect, more generally, arbitrary unimodal, symmetric distributions. The assumption can be further relaxed to detect the symmetry level of other distributions such as cyclic distributions (as shown in the updated manuscript) and possibly, other distributions and groups.
>
> 4 **The need for (and the quality of) the self-supervision and...**
>
> +-> The $\Theta$ network is designed for generalization to new datasets (meaning generalization to test datasets, if we are understanding you correclty) without requiring retraining or pre-training on these datasets. Regarding the reasoning about relying on pseudo-labels for predictions during testing: If we were to rely solely on pseudo-labels as predictions during the test phase, we would indeed have to re-run the pre-training phase on the test dataset. This approach contradicts the standard testing methodology, where the model should generalize to new data without training again. Additionally, using pseudo-labels as predictions would be problematic for smaller test datasets or individual input samples. For instance, if we only have a small set of 200 test samples with several classes in it, using these for neighbor computations would likely yield poor-quality neighbors. More critically, predicting the symmetry level of single inputs would be impractical, as the input would be its own neighbor, rendering the approach ineffective. In contrast, in our approach, once the $\Theta$ network is trained, it is capable of inferring the symmetry level of new, unseen inputs independently. The IE-AE's role is primarily during the training phase, where it aids in estimating the symmetry levels that are subsequently used to train the $\Theta$ network. After this training phase, the IE-AE becomes redundant and can be discarded. The trained $\Theta$ network is what we use for direct inference on new data, which is how we obtained the symmetry level prediction results for the experiments shown. In summary, the design of the $\Theta$ network is such that it eliminates the need for recalculating or relying on pseudo-labels during the testing phase, addressing in this way the concerns of efficiency and practicality. We updated the manuscript to include this nuance in our explanation (see Section 3. All changes in the revised manuscript are coloured in red).

---

> ### Author Response · Authors · 2023-11-23
> **Reviewer 9t9K**
>
> 5 **The way of calculating nearest neighbors... How are nearest-neighbors…**
>
> +-> The methodology for calculating nearest neighbors is indeed crucial for estimating the symmetries in the equivalence classes and is briefly mentioned in the manuscript. The explanation in the manuscript states: ``Let $N_{k,d}{=}N_k: X \rightarrow P(X)$ be a function that maps each input $x$ to the set $N_{k,d}(x)\subset X$ of $k$-neighbors around $x$ in the $G$-invariant latent space $Z_\mathrm{inv}$ as measured by some distance metric $d$." To put this idea in more detail: the set $N_k(x)$ consists of the $k$ elements $y$ of the dataset whose G-invariant embeddings $\eta(y)$ are nearest to the G-invariant embedding of the input $\eta(x)$, based on the chosen distance metric d. For instance, to compute $N_2(x)$, we first calculate the distance between every $\eta(y)$ in $\eta(X)$ and the embedding $\eta(x)$. Then, we select the two elements $y\in X$ that have the smallest distances in $\eta(X)$ to $\eta(x)$. We hope this answers the question, and have updated the manuscript to include a more explicit explanation of this computation (Section 3.4).
>
> 6 **Generating the pseudo-labels: Why is the estimator...**
>
> +-> The decision to use the method of moments estimator for generating pseudo-labels is motivated by its robustness to outliers, compared to other potential estimators. Let's consider the alternative of a maximum likelihood estimator (MLE) for the distribution of rotation angles. The MLE is the maximum angle observed in the nearest neighbors set $N_k(x)$. However, this approach is particularly sensitive to outliers. For example, if any element in $N_k(x)$ has an erroneously predicted high rotation angle, the pseudo-label for $x$ would be disproportionately influenced by this outlier. In contrast, the method of moments estimator in the absolute-valued distribution, which we employ, calculates the pseudo-label as two times the mean of all angles in $N_k(x)$. This averaging effect mitigates the impact of any occasional anomalous angles, as these outliers are diluted when calculating the mean. The previous reasoning is now included in the updated manuscript (see Appendix C).
>
> 7 **Have you tried to demonstrate the method on more realistic data?**
>
> +-> While the method promises to be applicable to other datasets such as the ones tested in the IE-AE work, in datasets with high intra-class variability e.g. CIFAR10, objects belonging to the same class may not share a G-invariant representation that connects them through a group action. In such cases, the standard IE-AE is unable to capture this relationship, and consequently, our method is also limited, as it relies on IE-AEs. We have included this discussion in the limitations section of our method (Section 6).
>
> ---
>
> We hope that these responses clarify your questions and concerns. Please let us know if you have any follow-up / additional questions.
>
> Best regards,
>
> The Authors

---

### Meta-Review · Area_Chair_gXsi · 2023-12-08

**Metareview:**

his work proposes a method for unsupervised learning of the symmetries of input samples.  Evaluation is performed on a version of MNIST in which different classes have different amounts of rotational variation.  The authors also prove a result giving a condition on when learning the distribution of symmetries is possible.

Many reviewers felt this work addressed a challenged and interesting problem and represented a novel advancement of previous work on invariant-equivariant autoencoders.  The experiments over SO(2) demonstrate the method can work as intended.  Moreover, the method rests on a solid theoretical foundation.  However, most reviewers felt that the work requires more empirical validation which is less limited than the current experiment which is on a synthetic dataset using only uniform distributions over SO(2).  The value of the method would be more clear if applied to an unaltered real world dataset.

**Justification For Why Not Higher Score:**

- limited experiments

**Justification For Why Not Lower Score:**

N/A

---

### Decision · Program_Chairs · 2024-01-16

Reject